# Flood risk assessment due to cyclone induced dike breaching on coastal areas of Bangladesh

Md Feroz Islam[1], Biswa Bhattacharya[2], Ioana Popescu[2]

[1]Copernicus Institute, Department of Environmental Sciences, Utrecht University, Utrecht, 3584 CB, the Netherlands
[2]IHE Delft Institute for Water Education, Delft, 2611 AX, the Netherlands

*Correspondence to*: M. F. Islam (m.f.islam@uu.nl)

**Abstract.** Bangladesh, one of the most disaster-prone countries in the world, has a dynamic delta with 123 polders protected by earthen dikes. Cyclone induced storm surges cause severe damages to these polders by overtopping and breaching of dikes. Nineteen major tropical storms hit the coast in last 50 years and the storm frequency is likely to increase due to climate change.

The present paper presents an investigation of the inundation pattern in a protected area behind dikes due to floods caused by storm surges and identifies possible critical locations of dike breaches. Polder 48 in the coastal region, also known as Kuakata, was selected as the study area. A HEC-RAS 1D-2D hydrodynamic model was developed to simulate inundation in the polder under different scenarios. Scenarios were developed by considering tidal variations, angle of cyclone at landfall, possible dike breach locations and the sea level rise due to climate change according to the fifth assessment report (AR5) of

Intergovernmental Panel on Climate Change (IPCC). Storm surge for a cyclone event of 1 in 25 year return period was considered for all the scenarios. The primary objective of this research was to present a methodology for identifying the critical location of dike breach, generating flood risk map (FRM) and probabilistic flood map (PFM) for the breaching of dike during a cyclone. The critical location of the dike breach among the chosen possible locations was identified by comparing the inundation extent and damage due to flood corresponding to the developed scenarios. A flood risk map (FRM) corresponding

to the breaching at the critical location was developed, which indicated that settlements adjacent to the canals in the polders were exposed to higher risk. A probabilistic flood map (PFM) was developed using the simulation results corresponding to the developed scenarios, which was used to recommend the need of appropriate land use zoning to minimize the vulnerability to flooding. The developed hydrodynamic model can be used to forecast inundation, to identify critical locations of the dike requiring maintenance and to study the effect of climate change on flood inundation in the study area.

The frequency and intensity of the cyclone around the world are likely to increase due to climate change which will require resource intensive improvement of existing or new protection works for the deltas. Identification and prioritising maintenance of critical location of dike breaching can potentially prevent a disaster. Non-structural tools such as land use zoning with the help of flood risk maps and probabilistic flood maps has the potential to reduce risk and damage. The method presented in this research can potentially be utilized for the deltas around the world to reduce vulnerability and flood risk due to dike breach

cause by cyclone induced storm surge.

## 1. Introduction

Bangladesh is the low lying delta of three major rivers: Ganges, Brahmaputra and Meghna. Eighty percent of the country's land is located below 10m AMSL (above mean sea level) (Heitzman and Worden, 1989) and it is formed of sediments carried by the above mentioned rivers. The population of Bangladesh was about 131.5 million by the year 2000 (World Bank, 2018) of which about 49% were living in the coastal zones (Neumann *et al.*, 2015). The coastal areas of Bangladesh are flooded frequently due to cyclone induced storm surges and occasionally due to high water levels in the rivers caused by heavy rainfall in the upstream catchments of Ganges, Brahmaputra and Meghna. The coast was hit by 5 severe cyclones between 1995 and 2010 causing flooding, huge damages and loss of life (Dasgupta *et al.*, 2014).

Bangladesh has 123 polders in the coastal area, each surrounded by earthen dikes, which are designed to protect the inland from flooding due to high tides. The existing crest level of these dikes are only adequate enough to protect the coastal area from cyclones with 5 to 12 year return periods (Islam *et al.*, 2013). These dikes usually get damaged and sometimes breached by tropical cyclones of high intensity, which causes flooding inside the protected areas, damages to properties and loss of life. For example, cyclone Sidr hit the coast of Bangladesh in 2007 affecting 8.9 million people, causing US$1.7 billion of damage (GOB, 2008; Dasgupta *et al.*, 2014). In 2009. Cyclone Aila affected 3.9 million people with an estimated damage of US$270 million (EMDAT, 2009).

Crest levels of the coastal dikes were recently designed for an event of 25 year return period under the Coastal Embankment Improvement Project (CEIP) (BWDB, 2013). A storm surge event of 25 year return period was considered in this study for the generation of different scenarios. Under the project CEIP the crest levels of the dikes were designed considering wave actions, astronomical tides and the required free-board. Raising the crest level was considered as the only mitigating measure. Various studies on the coastal areas of Bangladesh (e.g. Karim and Mimura, 2008; IWM, 2005; Azam *et al.*, 2004; Madsen and Jakobsen, 2004; CSPS, 1998; Flather, 1994) considered flooding only due to overtopping of the dikes during storm surges. Effect of breaching of the dikes due to piping and scouring on the landside during cyclones have not been studied. The coast of Bangladesh is frequently hit by severe cyclones (5 cyclones between 1995 and 2010, Dasgupta *et al.*, 2014). Bangladesh Water Development Board (BWDB) is responsible for the operation and maintenance of these dikes and lacks fund to conduct proper repairing of damaged dikes subsequent to any severe cyclone. As a result the dykes remain vulnerable to breaching. Identifying the critical location(s) of dike breach and prioritising the repairing of the critical location is likely to reduce the breaching possibility.

Moreover, non-structural measures for flood risk management such as land use zoning using the flood risk map (FRM) and probabilistic flood maps (PFM) to locate the vulnerable areas are currently unavailable for the coastal areas of Bangladesh. Flood zoning can be a useful risk mitigation measure as land use governs the exposure and may aggravate the hazard (Barredo and Engelen, 2010).

Furthermore, the intensity and frequency of these tropical cyclones are likely to increase in the future due to climate change. It is projected that by the year 2100, the frequency of the most intense cyclones will increase substantially and the intensity of

tropical cyclones will increase by 2 to 11% due to global warming (Knutson *et al.*, 2010). Flooding by tropical cyclones will also increase in the future as a result of sea level rise (SLR) (Woodruff *et al.*, 2013). SLR and sea surface temperature (SST) will affect the cyclone induced storm surge height in the Bay of Bengal (Karim and Mimura, 2008). With increasing SST, the storm surge height may increase from 21% to 49% and with SLR, the flood depth due to storm surges may increase by 30-
40% (Karim and Mimura, 2008). The land subsidence in the delta will exacerbate the effect of SLR. By the year 2100 the annual estimated damage due to tropical cyclones may increase by US$53 billion (Mendelsohn *et al.*, 2012).

At present, flood forecasting system is not available for the coastal region of Bangladesh. Bangladesh Water Development Board (BWDB), which is mandated to protect the area, does not have a clear picture about the inundation patterns corresponding to various climatic conditions. Moreover, identifying zones in the embankment critical to flooding in the polder
will help BWDB in prioritising their maintenance. This paper presents a methodology to identify the critical location of dike breach due to cyclones, generate flood risk map (FRM) and probabilistic flood map (PFM) due to breaching of the dike by cyclone induced storm surges. Different scenarios of storm surges were formulated by considering storms of different frequencies with varying tidal conditions, angle of cyclone at landfall and SLR. A cyclone event of 25 year return period was considered in this research. A coastal polder (Polder 48) of southern Bangladesh was selected as the study area.
With the effect of climate change, the frequency and intensity of the cyclone around the world will increase. The protective structures for the deltas around the world will require adjustments which will be resource intensive. Non-structural tools such as land use zoning with the help of flood risk maps and probabilistic flood maps has the potential to reduce risk and damage. Identification of critical location of breaching and intensifying the maintained effort for these locations can potentially prevent a disaster. Although coastal region of Bangladesh was selected as the case study area, the method presented in this research
can be utilized for the vulnerable deltas around the world.

## 2.  Study area

Polder 48, which was considered as the study area for this research, is surrounded by dikes and has a sea facing dike of about 20 km length on the southern side of the polder. The polder is located in the south western coast of Bangladesh Delta (Figure 1) stretching from $21^0 50'28''$ N $90^0 05'17''$ E to $21^0 50'06''$ N $90^0 14'14''$ E. The outline of the study area in Figure 1 (also in
Figure 3) depicts the dike alignment of the study area as well. The area is also known as Kuakata and it is in the administrative zone of Kalapara sub-district (Upazilla) of Patuakhali District. It has an area of 50.75 km$^2$ with 24,240 inhabitants according to the census of 2011 (BBS, 2012). Most of the inhabitants are farmers and fishermen (Nasreen *et al.*, 2013). Shrimp culture and tourism are also part of the economic activities. The land use is classified by the Ministry of Land, Bangladesh into the following four classes: rice fields, settlements, shrimp ponds and water bodies (river/canal). Climate and agricultural practices
of Kuakata is similar to the climate and agricultural practices of the country (Bangladesh). The average yearly rainfall in Kuakata is 2590mm (Climate-Data, 2016) and the annual average temperature is 25.9$^\circ$C (Climate-Data, 2016). Rabi (November to February), Kharif-I (March to May) and Kharif-II (June to October) are the three seasons for growing crops

(DAE, 2009). The elevation of 80% of the area is 1.55m PWD, the vertical datum established by Public Works Department of Bangladesh, which is 0.46m below the MSL. The land level surveys at different times have indicated that this polder is facing land subsidence issues. Brown and Nicholls (2015) reported the estimated mean subsidence rate of Ganges-Brahmaputra-Meghna (GBM) delta as 5.6 mm per year with an overall median of 2.9 mm per year.

5     The area was severely affected by recent storms Sidr, Aila and Mohasen in 2007, 2009 and 2013 respectively.  For example, during cyclone Sidr, 94 people died and 45% of the crops were lost (Ahamed, 2012).

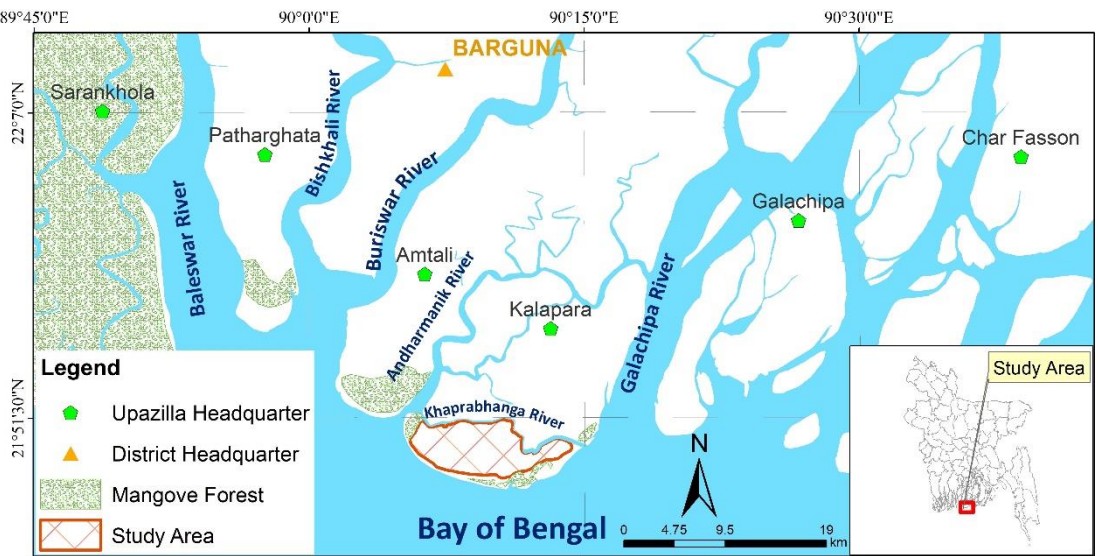

**Figure 1: Location map of the study area Polder 48 (Kuakata).**

Andharmanik, Galachipa and Khaprabhanga rivers are on the east, west and north of the study area respectively, whereas the Bay of Bengal is on the southern side of the study area (Figure 1). Galachipa River is the widest among the rivers surrounding the area. On the southern side, the study area has sea shore of 20 km width, which is partly protected by the mangrove forest

10     at several locations. There is a narrow sea beach in the south-western side of the area. The western part of the sea facing dike was overtopped during cyclone Sidr causing flood inside the polder (Hasegawa, 2008). The loss of livestock and food grains were such that it created partial deficiency of food in Kuakata (TANGO International, 2010). The average crest level of Polder 48 in the northern side is 4.5m PWD and in the southern side (sea facing side) is 6m PWD (Islam *et al.*, 2013). The existing embankments of 17 polders of the region, including Polder 48, were redesigned and rehabilitated during the first phase of

15     CEIP (Islam *et al.*, 2013). CEIP proposed a crest level of 7.36 m PWD for the dike of Polder 48 (Islam *et al.*, 2013).

## 3. Methodology

The methodology followed is presented in Figure 2 and described in the following sub-sections.

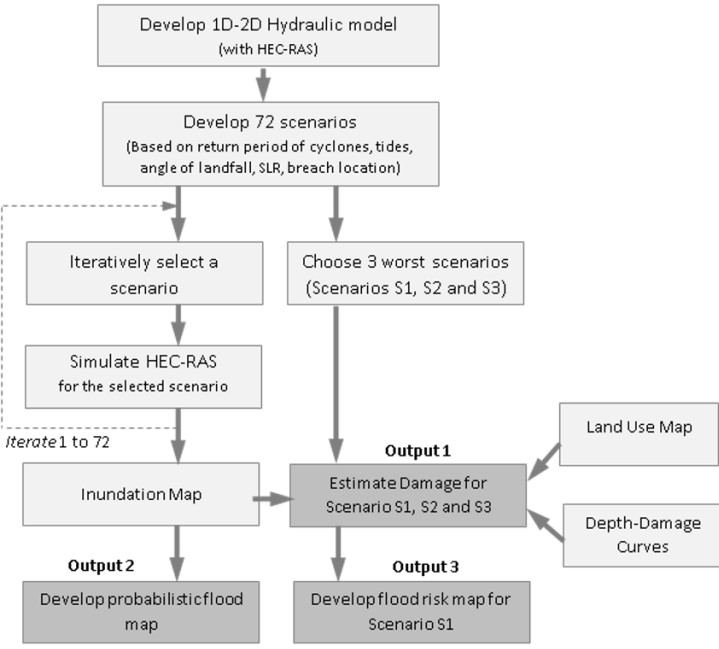

**Figure 2: Methodological approach followed in this study.**

### 3.1 Setting up of 1D-2D coupled model

In order to build a 1D-2D inundation model field measurements (land level surveys, observed water levels, canal alignments and cross sections of the river and canals) and information from remote sensing (satellite imagery) were gathered (Figure 3). Institute of Water Modelling (IWM) of Bangladesh collected hydraulic, hydrologic and land level data of the study area (along
5   with other polders) in the framework of the feasibility study of Coastal Embankment Improvement Project (CEIP). IWM has kindly provided the measured data for the study area.

The Digital Elevation Model (DEM) was generated by combining the land level surveys conducted by IWM and FINMAP. The land level survey of IWM (conducted in 2012) did not cover the whole study area. FINMAP conducted the topographic
10   survey of the study area in 1988 (MIWF, 1993). The differences in elevation between land surveys of IWM and FINMAP indicated the land subsidence. An average subsidence was computed, which was used to update the elevations of the FINMAP survey for the areas within Polder 48 for which survey data from IWM was not available. The combined DEM has a resolution of 50 m. The same DEM was used for the simulations of the year 2100 without any corrections for further subsidence. Subsidence of the coast in the past has been reported by Brown and Nicholls (2015) and was verified with the survey data
15   from IWM and FINMAP. Subsidence in the future may continue but in the absence of scientific studies it was not considered for the future scenarios in this research. It is noteworthy that if subsidence continues then the effect of the SLR may be increased and the results reported in this research should be treated as to some extent under-estimated values.

The bathymetry of the sea near the coast was collected from global bathymetric chart of ocean (GEBCO) (Smith and Sandwell, 1997). The land use data was collected from the Ministry of Land of Bangladesh. MODIS reflectance data was used for the analysis of previous flood events. The methodology and equations suggested by Hoque *et al*. (2007) were used to analyse the MODIS reflectance data to determine flood extents during previous flood events. The intention was to utilise the flood extent generated from MODIS reflectance for calibration of the hydraulic model. However, no flood image from MODIS was available during the simulation period.

The river analysis tool HEC-RAS (version 5.0) from the US Army Corps of Engineers was used to develop the 1D-2D coupled inundation model. The flow in the river was modelled in 1D whereas the flow over the floodplain was modelled in 2D. HEC-RAS 5.0 is a free tool which can simulate 1D, 2D and 1D-2D coupled models for steady and unsteady flow. The 2D module of HEC-RAS provides option to simulate flow of water either with the diffusion wave equation or with the full shallow water equation (St. Venant equation). The availability of irregular flexible mesh in HEC-RAS and the option for faster simulations led to the selection of HEC-RAS 5.0 as the modelling tool. Data utilized for developing the model and their sources are presented in Table 1.

**Table 1: The data used in developing the mathematical model and their sources.**
IWM, GEBCO and JSCE stand for Institute for Water Modelling, General Bathymetric Chart of the Oceans and Japan Society of Civil Engineers respectively.

| Component of the Model | Data collected and used | Source |
|---|---|---|
| 1D networks | Alignment | River network: IWM; The network on the sea side: Satellite image (Google Earth) |
| | Cross section | River: IWM; Bathymetry of the sea side: GEBCO |
| | Water Level | IWM |
| | Discharge | IWM |
| 2D Mesh | DEM | IWM and FINMAP |
| | Land Use | Ministry of Land |
| | Dike Alignment | IWM |
| 1D-2D coupled model | Crest Level of the existing dike | IWM |
| | Geometric properties of the dike | IWM |
| | Design crest level of the dike for future development | Islam *et al.*, 2013 |
| | Storm surge height | Azam *et al.,* 2004, Islam *et al.*, 2013 and Dasgupta *et al.*, 2014 |
| | Flood depths of previous events | JSCE |

The 1D section of the model was developed and calibrated using the information shared by IWM. The 1D part of the developed model was calibrated for non-flood conditions as measured discharge/ water level data during a cyclone event was unavailable. The model was simulated using discharge as the west boundary and water level as the east boundary conditions (Figure 3). The calibrated 1D model was then coupled with the 2D model of flow over the floodplain using the DEM of the study area.

5 For the 1D-2D inundation model, a computational mesh with flexible shape was developed in HEC-RAS (Figure 3). HEC-RAS generates mesh with irregular shapes. The rectangular cells of the developed 2D mesh had a resolution of 25 m and the non-rectangular cells had areas ranging from 625 to 1282 square meters. The roughness coefficient (Manning's $n$ varying from 0.025 to 0.05) was provided according to the landuse of each cell. A sensitivity analysis as suggested by Hall *et al.* (2005) was carried out by varying Manning's roughness coefficient $n$ before the calibration of the 2D inundation model.

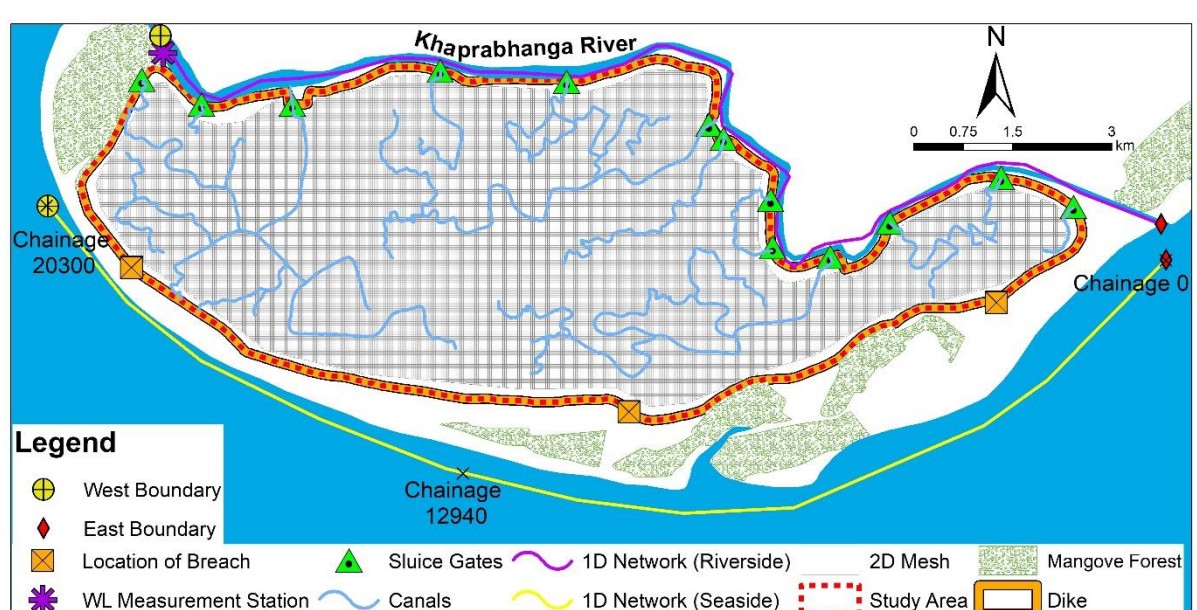

**Figure 3: Schematic diagram of the study area with location of control structures and gauges and the considered breach locations.**

Building and calibrating the 1D model was the preliminary step for developing the 1D-2D coupled model. The water bodies surrounding the study area were included in the 1D model. The study area has Khaprabhanga River on the northern side and

15 sea on the southern side (Figure 3). The connection of Khaprabhanga River with other rivers was not considered in the model. This was due to the fact that storm surges are observed during the pre- or post-monsoon periods whereas fluvial floods are observed during the monsoon. Flow through rivers did not play a major role during the previous cyclones. The western and eastern side of the embankment have Mangrove Forests between the rivers and the embankment (Figure 3).

For the river, the surveyed cross sections were used in the 1D model (Figure 4). The storm surge on the sea was conceptualised

20 as a water surface profile in a 1D channel on the southern side of the study area (Figure 3). The GEBCO bathymetry (Figure 5) was used for the channel. An alternative was to develop a 2D model for the coastal hydrodynamics. However, as the coast

of Bangladesh is flat and shallow a large area of the sea would have been included in the model. As the focus was on studying the inundation in Polder 48 and not the coastal sea we followed a simpler representation of the storm surges using a 1D model. The synthetic water level data for boundaries of the model were generated by following the tidal water level pattern and the storm surge height considered for the all scenarios (Table 2). The water surface profile corresponding to each scenario (Table 2, discussed in Section 3.2) was considered as the profile in the 1D model of the sea side (Figure 6).

The dense canal network of 122 km, inside the study area, is connected with the Khaprabhanga River, which regulates the in- and out flow into the river network through a system of 13 control structures. The regulators remain closed during cyclones making the canal network isolated. Therefore, the canal network inside the polder was not included in the 1D model. However, the simulation of the overland flow consequent to breaching of the dike will be affected by the canal geometry and therefore, the wider and larger canals were included in the DEM.

The geometry and propagation of the breach of the dike depends primarily on the storm surge height, angle of landfall, soil properties and wave action. The coastal embankments of Bangladesh are usually earthen. The geometrical properties of the breaching of the dike and the time required for breaching were calculated following the instructions of US Bureau of Reclamation. An *S*-curve was used for breach propagation with time (Oumeraci, 2006). As the geometry of the breach is not independent, it was not considered as a parameter for scenario development.

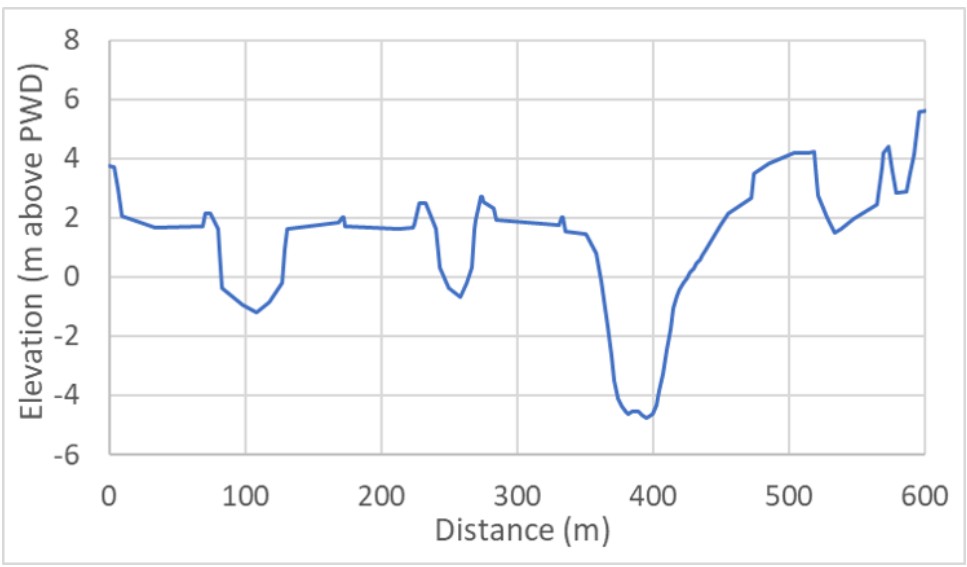

**Figure 4: A typical cross section of the Khaprabhanga River**

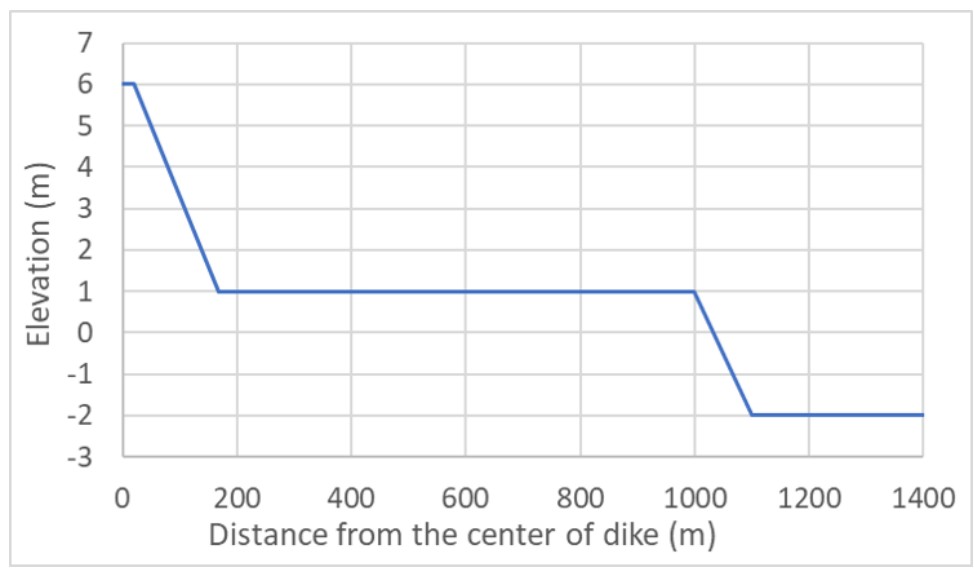

**Figure 5: Cross section for the 1D network on the sea side**

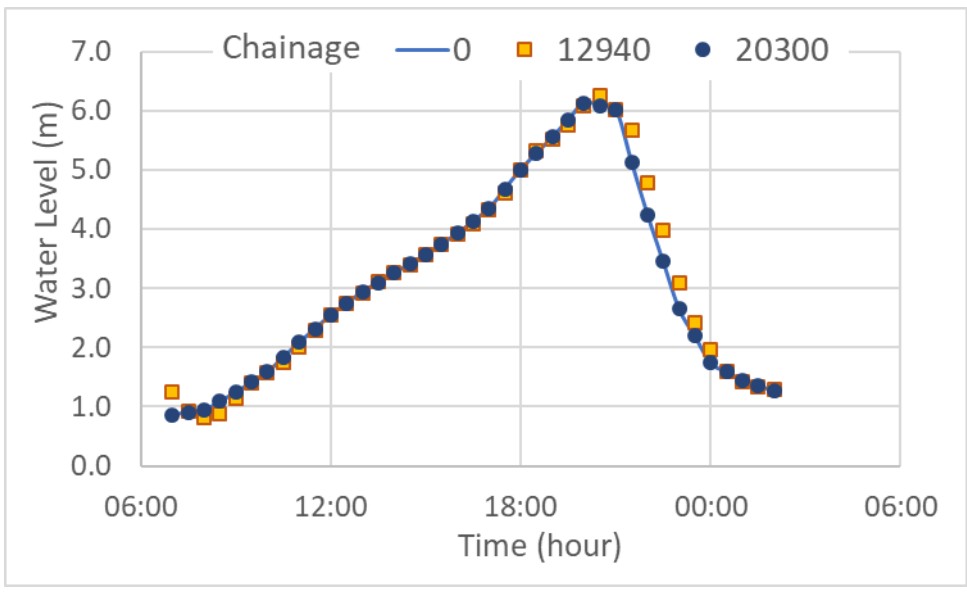

**Figure 6: Variation of water level at three locations (chainage 0, 12940 and 20300) along the 1D channel on the sea side according to a specific Scenario (out of 72 Scenario).**

In order to ensure model stability a maximum spacing between the computational points was imposed and computed using Samuels' formula (1989) presented in Eq. (1):

5    $\Delta x \leq 0.15 * D/So$ (1)

where, $\Delta x$ is the spacing between the computational points, $D$ is the average bank full depth of the channel and $S_0$ is the average slope of the channel. The maximum spacing between cross sections was calculated as 300 m. The river had steeper bed slope than the long shore slope of the sea bathymetry requiring smaller $\Delta x$ to ensure stability and same $\Delta x$ will reduce instability of the foreshore as well.

As suggested by Fromm (1961) the Courant number was kept less than or equal to 1.0 to maintain the stability of the numerical model by controlling the time step. The Courant number was calculated using the following Eq. (2):

$$Cr = V * \Delta t / \Delta x \qquad\qquad\qquad (2)$$

where $Cr$ is Courant number, $V$ is velocity, $\Delta t$ is the time step and $\Delta x$ is the spacing between the cross sections.

### 3.2 Cyclonic scenarios considered

Different scenarios were developed considering the probability of occurrence of cyclones, the angle of landfall, SLR due to climate change, diurnal, semi-diurnal and seasonal variation of tides, locations of breaching of the dike and geometrical properties of the breach.

*Frequency of cyclone*: A cyclone of 1 in 25 year return period was considered for all the scenarios as this is used as the design criteria for the dikes (BWDB, 2013). Nineteen previous cyclones for different tidal conditions were simulated by IWM using

a 2D model for the Bay of Bengal. A statistical analysis was conducted using these model results to generate the storm surge height corresponding to a cyclone of 25 year return period (Islam *et al.,* 2013). Due to lack of data, change in the probability of occurrence of the cyclone in the future was not considered.

*Angle of landfall*: The angle of landfall affects the height of storm surges. The storm surge height increases with angle of the storm to the coastline (Azam *et al.,* 2004). Angle of attack governs the wind speed which is one of the parameters for the height

of cyclone induced storm surges (Azam *et al.,* 2013).

*Tides*: The difference between the storm surge at high tide and low tide is 1.2m for the study area (Azam *et al.,* 2004). The average seasonal variation of the tidal range is 1.3m.

*Sea level rise*: The coast of Bangladesh may be severely affected by SLR and one fourth of the land may be lost due to SLR by 2100, which will directly affect 3 million people (Ericson et al., 2005). IPCC published their 5th assessment report (AR5)

in 2013. Among the scenarios considered in AR5, RCP (representative concentration pathways) 2.6 is the most optimistic one and RCP 8.5 is the worst considering the carbon emission, rise in temperature and SLR. The mean SLR at the end of 21st century is estimated to be 0.4m, 0.47m, 0.48m and 0.63m for RCP 2.6, RC 4.5, RCP 6.0 and RCO 8.5 respectively (Stocker *et al.*, 2013). For this study, RCP 8.5 with SLR of 0.63 m was considered for developing the scenarios.

*Location of breach*: The sections of the sea facing dike of the study area protected by mangrove forest, sand dunes and wide

beach are least likely to be breached due to storm surges. The study considered breach locations with least protection. The considered locations for dike breach as well as the mangrove forest around the study area are shown in Figure 3.

A scenario matrix consisting of 72 scenarios were generated by combining different phases of tides, angle of landfall, SLR and breach locations (Table 2). Single breach was considered for all the scenario. The highest storm surge height among all

the developed scenarios was 7.2 m PWD considering the angle of landfall as $230^0$, high tidal phase during spring tides, SLR and dike breaching at any of chosen locations. The highest storm surge height as the boundary condition with breaching at the western, central and eastern parts of the dike were considered as the worst case scenarios and were denoted as Scenario S1, S2 and S3 respectively (Figure 2). Flooding due to overtopping of the dikes was not considered as the crest level (7.36 m PWD) was higher than the highest storm surge height (7.2 m PWD).

**Table 2: Storm surge heights corresponding to different Scenarios considered**

| Angle of Landfall | Tidal Variation | | East | | West | | Central | |
|---|---|---|---|---|---|---|---|---|
| | | | With SLR | Without SLR | With SLR | Without SLR | With SLR | Without SLR |
| | | | | | Storm Surge Heights | | | |
| 200 | High Tide | Spring tide | 4.06 | 3.38 | 4.06 | 3.38 | 4.06 | 3.38 |
| | | Neap tide | 2.77 | 2.09 | 2.77 | 2.09 | 2.77 | 2.09 |
| | Low Tide | Spring tide | 2.83 | 2.15 | 2.83 | 2.15 | 2.83 | 2.15 |
| | | Neap tide | 1.54 | 0.86 | 1.54 | 0.86 | 1.54 | 0.86 |
| 215 | High Tide | Spring tide | 6.16 | 5.48 | 6.16 | 5.48 | 6.16 | 5.48 |
| | | Neap tide | 4.86 | 4.18 | 4.86 | 4.18 | 4.86 | 4.18 |
| | Low Tide | Spring tide | 4.93 | 4.25 | 4.93 | 4.25 | 4.93 | 4.25 |
| | | Neap tide | 3.63 | 2.95 | 3.63 | 2.95 | 3.63 | 2.95 |
| 230 | High Tide | Spring tide | **7.20** | 6.52 | **7.20** | 6.52 | **7.20** | 6.52 |
| | | Neap tide | 5.91 | 5.23 | 5.91 | 5.23 | 5.91 | 5.23 |
| | Low Tide | Spring tide | 5.97 | 5.29 | 5.97 | 5.29 | 5.97 | 5.29 |
| | | Neap tide | 4.68 | 3.99 | 4.68 | 3.99 | 4.68 | 3.99 |

To identify the critical location of breaching, results of the scenarios simulated with HEC-RAS corresponding to the three worst case scenarios S1, S2 and S3 were compared based on the total area flooded and estimated damage due to flooding. Using the calculated damage and occurrence probability of the event, a risk map was generated for the critical locations of the sea facing dike. A probabilistic flood map (PFM) was generated from the flood maps of the 72 scenarios (Table 2; Output 2 in Figure 2). As the storm surge height suggested by Islam *et al.* (2013) corresponds to an event of 25 year return period, the PFM generated in this study corresponds to 1 in 25 year return period.

**3.3 Estimation of damage due to floods**

A comprehensive damage calculation should involve both direct and indirect damages due to floods (Büchele *et al.*, 2006). Direct damage is caused by physical contact of properties and human beings with flood water. Indirect damage is caused by interruption of services, production and transportation and degradation of health due to floods. Due to lack of data, only the direct damage to properties were calculated for the study area. The damage was considered as a function of flood depth. The land use of the study area was classified by the Ministry of Land of Bangladesh as settlements, rice fields, shrimp ponds and water bodies (rivers/canals). Only the tangible damage was considered and no environmental damage was calculated. Damages to the canal network was not considered. The damage in a flood event was calculated using Eq. (3).

$$D = \left(\sum_{i=0}^{n} x_i \times f(x_i)\right) \times A_i \qquad (3)$$

Where, $D$ = total direct tangible damage in a flood event, $n$ = total number of computational cells within the flooded area, $x_i$ = flood depth of cell $i$, $f(x_i)$ = damage function for the land use of the flooded cell $i$ and $A_i$ = area of cell $i$.

Depth-damage curves for different land classes for the study area were developed by adapting depth-damage curves found in the literature (Figure 7). Reese *et al.* (2010) calculated flood damage as a percentage of the property value of buildings categorised based on the construction material. The buildings of the study area are primarily built of timber due to its low cost and easy availability. The depth-damage curve suggested by Reese *et al.* (2010) for buildings made of timber was used as a basis for generating the depth-damage curve for the settlements (residential area). Simple Action for the Environment (SAFE) carried out a research on the average value of properties in rural areas of Bangladesh (SAFE, 2011). This property values were used to update the damage values used by Reese *et al.* (2010). Muktadir and Hasan (1985) reported that rural houses of Bangladesh are built with a large courtyard and as a result houses have a lot of open and unoccupied space around buildings. The damage curve considered for the residential area was used for the damage to the buildings and not for the courtyard. Moreover, the satellite image of the area also indicated that about half of the settlement was without buildings. Satellite image from google earth was used for analysis. The satellite image of the area was downloaded and georeferenced. Then, the areas for buildings and open areas for households were manually calculated using ARCGIS. Therefore, 50% of the settlement area was considered with no damages.

The cultivation of rice involves flooding the rice field with water up to a few cm. However, if the height of water increases and the rice plant goes under water then the productivity decreases. The damage to rice plants also depends upon the flood duration. If the rice plant is continuously under water for more than 2-3 days then the damage can be up to 80% (Chau *et al.*, 2014). The simplified (with regards to flow velocity and flood duration) depth-damage curve for rice fields suggested by Chau *et al.* (2014) was used in this study (Figure 7).

Shrimp ponds are surrounded by embankments so that there is no damage to shrimp ponds till the flood level crosses the embankment level. However, when the flood level is higher than the embankment level shrimps escape causing a loss of the total investment. To take this into account, the investment made by farmers was assessed using a study conducted by Fatema

*et al.* (2011). According to the study the investment for shrimp pond in the study area was about € 0.09/ m². Based on the practices in the study area the banks of the shrimp ponds were considered as 2 meter above the adjacent land and the depth-damage curve (Figure 7) was modified accordingly. The adapted depth-damage curves are obviously simplistic ones.

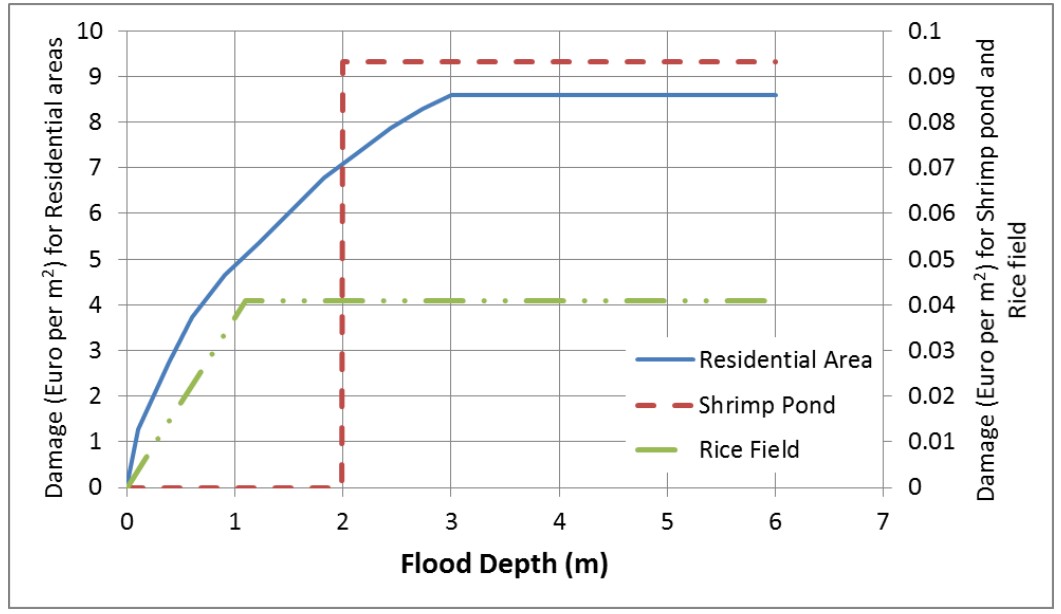

**Figure 7: Depth-damage curves for different land use classes.**

The damage calculations were carried out using ArcGIS. The simulated flood depth and landuse for each grid cell was used as the input and the damage in each grid cell was computed by using the depth-damage curve corresponding to that landuse. The damage for each scenario was estimated using this procedure.

### 3.4 Calculation of flood risk and generation of risk map

Flood risk assessment is an essential part of risk management. Spatial distribution of risk and areas requiring mitigation measures can be identified from flood risk maps. To examine the spatial variation of risk, flood risk analysis was carried out and a risk map was generated considering dike breaching at the critical location. Van Manen and Brinkhuis (2005) and Klijn (2009) under FLOOD*site* project carried out research to quantify the flood risk for the polders in The Netherlands for dike failure defining the risk as a product of the occurrence probability of the event and the consequences which was defined by Helm, (1996). Eq. (4) was used to calculate the risk due to flooding:

$$R = P_F * S \tag{4}$$

Where, $R$ = risk, $P_F$ = probability of occurrence of the flood hazard and $S$ = consequences

The exceedance probability (return period) of the cyclone induced storm surge was used as the probability of occurrence of the hazard. The probability of flooding within a protected area is not the same as the probability of the hazard and depends also upon the probability of failure of the dike. It is a difficult probability to compute as the probability of dike failure also

depends upon the dike maintenance, information about which were not available. Here we have assumed that the probability of occurrence of the hazard and the probability of failure of dike are the same.

## 3.5 Probabilistic flood map

Purvis *et al.* (2008) stated that the risk assessment for the most probable scenario cannot take into account the impact of the
scenario of low probability, stressing the necessity of a probabilistic risk analysis. The equation suggested by Purvis *et al.* (2008) for probabilistic risk analysis was adjusted and used for this research to calculate the probability of flooding of each cell and is presented below in Eq. (5):

$$P_{i(i=1\ to N)} = \frac{\sum_{j=1}^{M} F_{ij} \times P_{fj}}{\sum_{j=1}^{M} P_{fj}} \ and\ F_{ij} = \begin{cases} 1, & if\ flooded \\ 0, & if\ dry \end{cases} \tag{5}$$

Where, $P_i$ is probability of flooding at cell $i$, $P_{fj}$ is the probability of reaching a certain storm surge level in simulation number $j$, $F_{ij}$ is the binary value indicating if the cell $i$ is flooded or not in simulation $j$, $j$ =1,2,3, .., $M$ where $M$ is the number of Scenarios considered (=72), $i$ =1,2,3,…, $N$ are the computational grid cells on the polder area and $N$ is the number of cells. Equation (5) was used in this study to calculate the probability of flooding at each cell. The probabilistic flood map (PFM) was calculated using the results of all the scenarios.

## 4. Results and discussion

The developed 1D-2D modelled for the present study was calibrated for the 1D part by comparing the observed and simulated values for discharge and water level. The corresponding performance indicators used for evaluation were coefficient of determination ($R^2$), root mean square error (RMSE) and mean absolute error (MAE) for which values of 0.98, 2.15 m$^3$/s and 1.68 m$^3$/s respectively were obtained for discharge; and 0.98, 0.09 m and 0.08 m respectively for water level. The average
value of the discharge and water level for the considered simulation period was 5.68 m$^3$/s and 0.82 m respectively. The period of simulation for calibration coincided with the surges corresponding to cyclone Sidr (from November 14 to November 17, 2007). The simulation results indicates that the dike facing the seaside was overtopped and the area inside the polder was inundated. This conclusion is in line with the survey conducted by Japan Society of Civil Engineers (JSCE) (Hasegawa, 2008). The coupled 1D-2D model has not been calibrated because there were no flood maps showing flood extents available for recent
cyclones. However, the 2D part of the model was pseudo-calibrated considering MODIS reflectance data. Such data was used in order to analyse the inundation extent, though this also posed considerable challenges due to the cloud coverage during the cyclones. The survey conducted by JSCE after cyclone Sidr was collected to investigate the flood extent and depth, which provided flood depth for only one location inside the study area. This location was used for the calibration of 2D model. The difference between the reported and the simulated flood depth was 4.5%. Prior to the calibration of the 2D model, sensitivity
analysis was carried out regarding the roughness coefficient (Manning's *n*). The analysis indicated that the inundation model

is not highly sensitive to the roughness coefficient and the areas of low flows (locations furthest from the dike breach) are most sensitive. The sensitivity analysis was done for the breaching on the western part of the dike only. It was considered that the breaching at other locations will have similar effects as the area inside the polder is flat and low lying with mostly farmlands near the dike.

5   This 1D-2D model, which had limited calibration points, was further used in simulating the developed scenarios. The simulated results were used to analyse flood depth, extent and damages due to flood. The FRM and the PFM were generated based on flood results of the model.

## 4.1 Inundation corresponding to three worst case Scenarios

Among the simulated scenarios, the results of three worst case scenarios (Scenario S1, S2 and S3) were compared for
10   identifying the critical location of breaching. The corresponding flood maps for the worst case scenarios are presented in Figure 8.

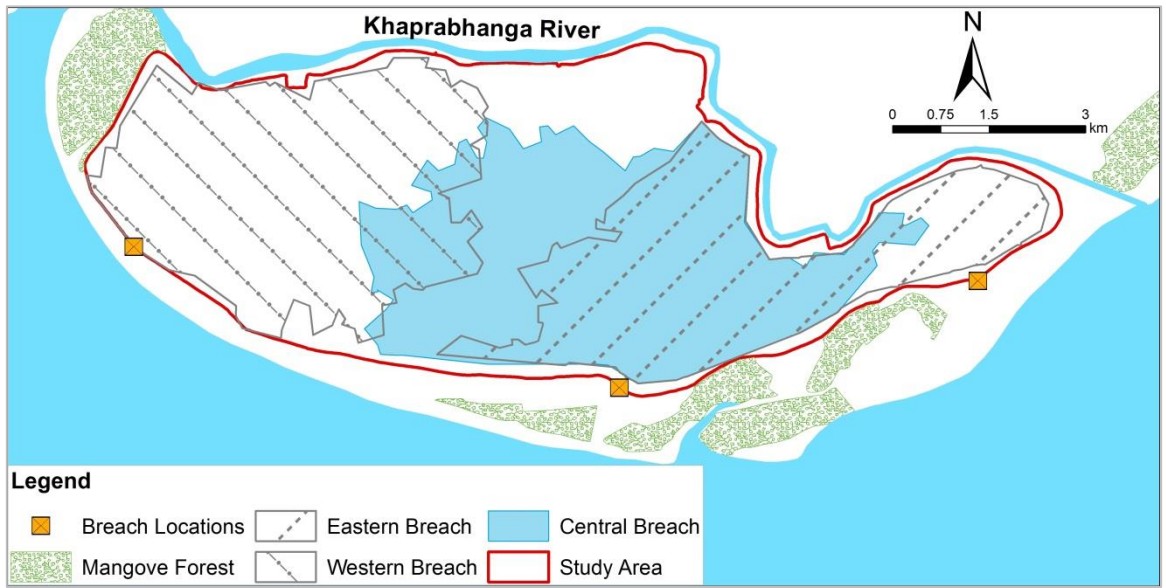

**Figure 8: Flood extent corresponding to three worst case scenarios of dike breaching in the central, eastern and western section of the dike.**

Flood extents corresponding to all different scenarios presented in Table 1 were compared to understand the effect of SLR, diurnal and seasonal tidal variation and angle of cyclone at landfall. The flood extents of different scenarios considering the breaching at central part of the sea facing dike is presented in Figure 9.

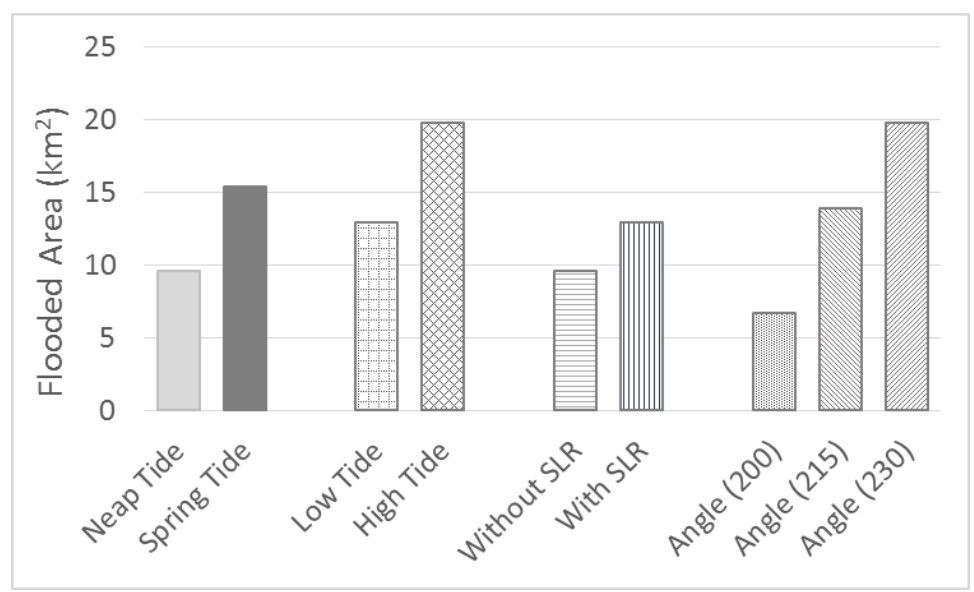

**Figure 9: Comparison of flooded areas corresponding to different Scenarios considering the breaching at central part of the sea facing dike.**

Moreover different land classes were considered while computing the flood extent for the three worst case scenarios (Table 3). The analysis of the flood extent for different flood depths, based on the considered land uses are presented in Figure 10. The highest storm surge height among all the developed scenarios were 7.2 m PWD (Table 2). This storm surge height with breaching at the western, central and eastern parts of the dike were considered as the worst case scenarios and were denoted

5   as Scenario S1, S2 and S3 respectively.

**Table 3: Flooded areas of different land classes corresponding to the three worst case Scenarios.**

| Land Classes | Flooded Area (km²) | | |
|---|---|---|---|
| | Scenario S1 | Scenario S2 | Scenario S3 |
| Rice fields | 15.3 | 16.4 | 12.3 |
| Settlements | 3.1 | 3.1 | 2.1 |
| Shrimp ponds | 0.2 | 0.1 | 0.1 |
| Canals | 1.2 | 1.7 | 1.2 |
| Total | 19.8 | 21.2 | 15.8 |

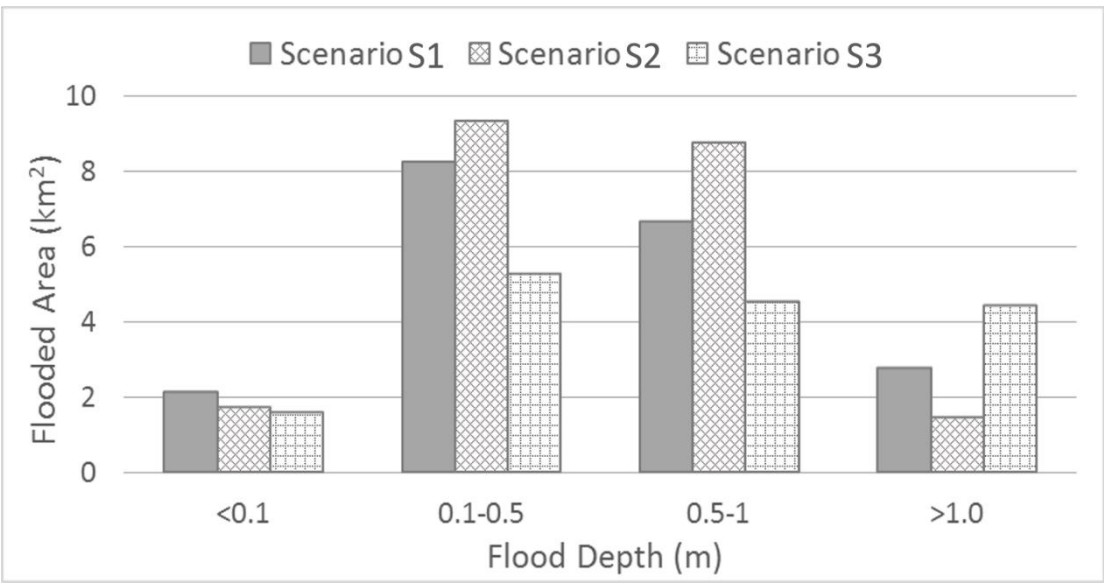

**Figure 10: Flooded areas for different ranges of flood depths corresponding to different Scenarios.**

## 4.2 Comparison of calculated damages

The damage due to flooding was calculated using the depth-damage curves for different land classes. The calculated damages for different land classes and damages for different flood depths corresponding to the three worst case scenarios are presented in Table 4 and Figure 11.

**Table 4: Calculated flood damages for different land classes corresponding to different scenarios.**

| Land Classes | Estimated flood damage (million Euros) | | |
|---|---|---|---|
| | Scenario S1 | Scenario S2 | Scenario S3 |
| Rice Fields | 0.4 | 0.4 | 0.3 |
| Settlements | 10.3 | 10.3 | 8.3 |
| Shrimp Ponds | 0.0 | 0.0 | 0.0 |
| Total | 10.7 | 10.7 | 8.7 |

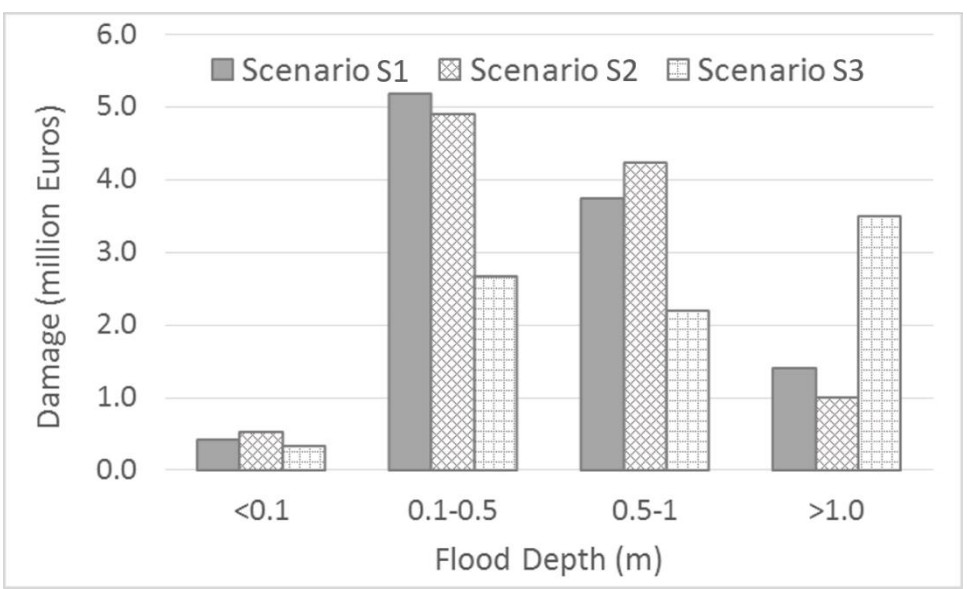

**Figure 11: Variation of estimated flood damages with varying ranges of flood depths corresponding to Scenario S1, S2 and S3.**

Figure 10 and Figure 11 corresponds to the flooded area and damage due to different range of inundation respectively. The flood area and damage were highest for the inundation depth of 0.5 m to 1.0 m.

## 4.3 Risk map for the worst case scenario

The flood risk map for the scenario with the critical location of breaching of dike is presented in Figure 12. The risk map presents the assessed risk of flooding due to breaching at critical locations of the dike. Comparison of flooded area and damage due to flood for the three worst case scenarios lead to the identification of Scenario S1 as the critical location of breaching. The identification of the critical location of breaching is described in section 4.5.

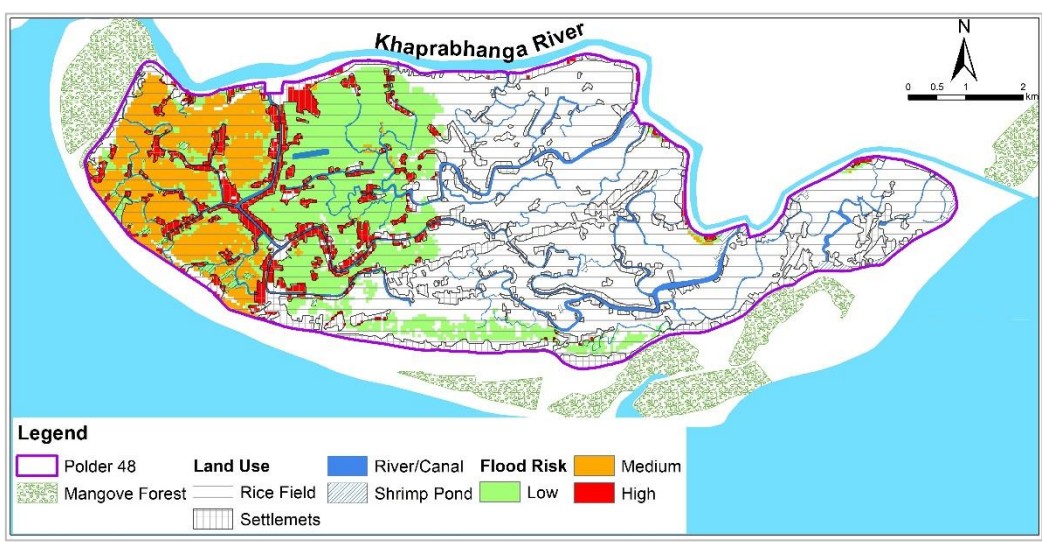

**Figure 12: Flood risk map corresponding to the dike breach at the critical location of the dike. The following three classes of risk are shown: high, medium and low. The considered four landuses are shown as well.**

## 4.4 Probabilistic flood map

Although the inundation maps are widely used for spatial planning and flood mitigation measures, the uncertainty of mathematical modelling affects the output of inundation maps (Alfonso *et al.*, 2016). In order to account for uncertainty, probabilistic flood maps are suggested to be used (Domeneghetti *et. al.*, 2013). The probabilistic flood map was calculated
5   from the inundation maps corresponding to the 72 scenarios considered in the study. Probabilistic flood maps were calculated for a threshold of flood depth greater than 0.5 m. The developed damage curves suggest that the damage for flood depth below 0.5 m is minimal. Moreover, considering the widely accepted ´living with floods´ philosophy in Bangladesh a threshold of 0.5m was adopted. This threshold was used in developing the PMFs. This threshold was not considered while the estimation of damage due to flood was conducted. The calculated probabilistic flood maps are presented in Figure 13. The probabilistic
10   flood map indicates the likelihood of being flooded. This will assist the planning for future land use zoning, which can be used to restrict further developments in the floodplains.

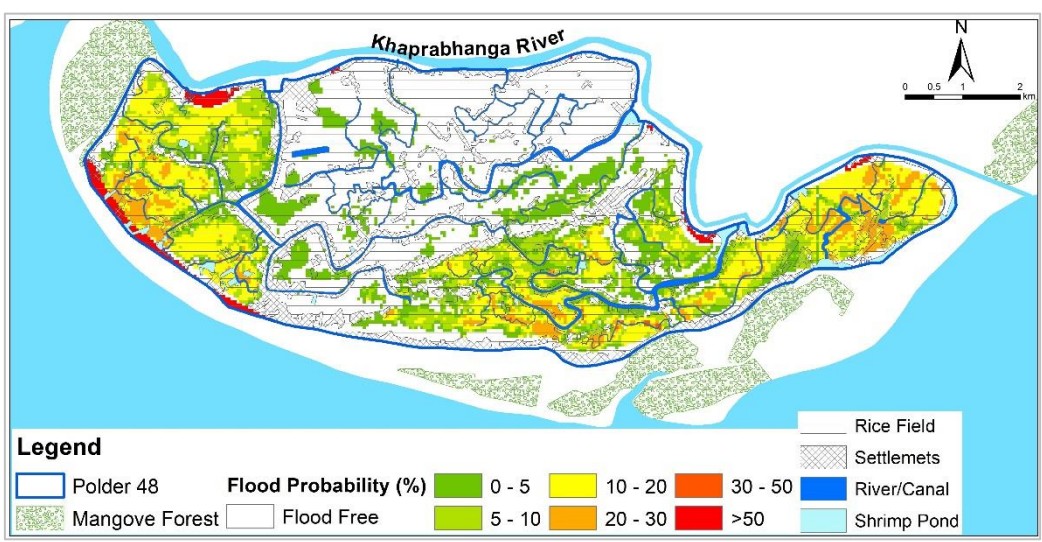

**Figure 13: Probabilistic flood map of the study area. Varying colours indicate probabilities of obtaining flood depths more than 0.5m.**

### 4.5 Discussion on the results

The flood extent for the simulated result of 72 scenarios were compared. Flood extent varies for different scenarios with different conditions such as the daily (high and low tide) and biweekly (spring and neap tide) tidal variation, sea level rise and angle of landfall (Figure 9).

5   Three worst case scenarios (Scenario S1, S2 and S3) were compared by generating flood maps, calculating total flooded areas and total damages. The flood maps for Scenario S1, S2 and S3 (Figure 8) demonstrated that a large area was flooded for all the breach locations. At least 25% of the total area of Polder 48 was inundated for the three scenarios (Table 3). In the case of all three considered scenarios, the inundation area having flood depths from 0.5 to 1.0 m, was larger than the inundation areas with other flood depths (Figure 10). The inundation area with flood depths more than 1m was largest for Scenario S3, due to

10  the depressions close to the dikes (Figure 10). The rice fields were flooded most while the shrimp ponds were flooded least in all the scenarios (Table 3).

Flood risk was quantified with damage due to floods (negative consequences) and probability of occurrence. The total estimated damages due to flooding for Scenario S1, S2 and S3 were 10.7, 10.6 and 8.6 million Euros, respectively (Table 4). For all the scenarios, a 1-in-25 year cyclone event was considered. The damage to the settlements was greater than other land

15  classes for all the scenarios (Table 4). Rice fields were flooded most but they did not experience the highest damage compared to other landuse classes (Table 3 and Table 4). This can be explained by the high damages in settlements compared to rice fields (Table 4). The damage to crops depends on the flood depth, duration and overland flow velocity. For simplification, only the damage related to flood depth was used. As the probability of cyclones were considered the same for all the scenarios, the calculated damage governed the estimated flood risk, i.e., higher damage to the settlements translated as higher risk of

flooding. The primary economic activity of the inhabitants of the study area is farming (BBS, 2011) and most of the inhabitants are poor (with a poverty rate of 0.628) (Alamgir *et al.*, 2018). Even though the estimated damage and risk of flooding to crops were much less compared to other land uses, it will affect the people living in the study area most as they depend on the farming of rice for their livelihood (Nasreen *et al.*, 2013). Hasan *et al.* (2004) found out that the dependence on fishing (in the sea) by

the inhabitants of Polder 48 are increasing due to loss of crops by flood, loss of productivity, lack of jobs and poverty. Fishing in the coastal region of Bangladesh yields lower economic returns leading to enhanced poverty (Hasan *et al.*, 2004).

The damage was maximum with flood depths 0.1 to 0.5m for all the scenarios (Figure 11). The damage due to inundation less than 0.1m was small and insignificant. The damage is a function of flood depth but the unit is per unit area(per m$^2$).Therefore, if the flood extent for higher depth is lower, the damage due to flood might be lower even though  the flood damage increases

significantly for inundation more than 0.5m according to the developed depth damage curves (Figure 7).

Generated PFM indicated that the areas adjacent to the dike facing the seaside have higher probability of flooding and the rice fields are more prone to flooding (Figure 13). Moreover, the areas protected by mangrove forest might also be flooded if the unprotected location of the dike is breached (Figure 13), stressing the importance of proper maintenance of the dike everywhere.

The damage due to flooding was maximum for Scenario S1 which results in higher risk of flooding for scenario S1. Total flooded area for settlements of Scenario S1 was lower than Scenario S2 (Table 3) but the estimated damage for settlements of Scenario S1 was more than Scenario S2 (Table 4). This indicates that the settlements for Scenario S1 were exposed to greater flood depth and higher risk of flooding than Scenario S2. Furthermore, Scenario S1 had similar total damage due to flood with lower flood extent than Scenario S2 (Table 3 and Table 4). Considering these facts, Scenario S1 was selected as the worst case

Scenario and breaching at the western part of the sea facing dike was identified as the critical location for breaching during cyclone.

The scenarios with the effect of climate change (sea level rise) had more damage compared to the scenarios without climate change. Scenarios S1, S2 and S3 were associated with the highest storm surge height. With the same set of conditions without climate change (sea level rise) the storm surge height was 6.52 m PWD (Table 2). The damage corresponding to breaching of

eastern, central and western locations of the dike due to the storm surge without climate change impact were 23.3%, 20.5% and 21.7% respectively lower than the damages with the climate change. The corresponding values for the flood extent were 30.1%, 21.67% and 27.21% respectively lower than the flood extent areas with the climate change impact.

The probability of occurrence of the storm surge and damage caused by inundation were taken into consideration for the risk calculation. In case of breaching of the dike, the probability of flooding was considered the same as the probability of

occurrence of storm surge. The depicted risk map (Figure 12) shows the areas adjacent to the dike breach is at higher risk and the risk reduces as the flood propagates towards east.

Canals are used as a mode of transportation by the inhabitants of the area. Most of the economic activities and residential areas are by the canals. The risk analysis show that the areas at highest risk are the settlements by the canals (Figure 12). Therefore, although canals plays a crucial role in the economy and social life of the area, they also increase the risk of flooding and probability of higher damage to the adjacent areas.

Land use plan plays an important role in reduction of vulnerability to disasters (Burby, 1998). Probabilistic flood maps (PFM) can be used for land use planning (Alfonso *et al.,* 2016). For better understanding of the area at risk of flooding due to breaching of dike, probabilistic flood maps (PFM) were generated for the study area (Figure 12 and Figure 13). The results of 72 Scenarios from Scenario matrix was used for calculation of PFM. The areas adjacent to the sea dikes had higher probability of flooding due to breaching of dike for both PFMs. The areas inland had lower probability of flooding. Existing land use indicates that
the areas with lower probability of flooding are mostly rice fields (Figure 12 and Figure 13). Land use zoning and management using the PMF can reduce the vulnerably.

## 5. Conclusions

A 1D-2D coupled model was developed to investigate the inundation pattern inside a polder due to breaching of a dike by cyclone induced storm surges. Different scenarios were formulated and simulated using a 1D-2D coupled model. The results
of these simulations were used to calculate the total flooded area and damage due to flooding. Simulated results of three worst case scenarios S1, S2 and S3 were compared based on the total flooded area and estimated damage. The comparison led to the identification of the critical location of dike breaching during a cyclone. The flood risk map and probabilistic flood map were generated for the dike breaching during a storm surge using the results of the developed scenarios to identify the areas at higher risk and higher probability of flooding.

Flood inundation for the three worst case scenarios S1, S2 and S3 indicated that the maximum flooded area was obtained corresponding to the breaching of the central part of the sea facing dike. For all three breaching locations most of the flooded area had flood depth of 0.1 to 0.5 m. The highest depth was obtained corresponding to scenario S2 (breaching in the central part). The damage for scenarios S1 (breaching in the western part) and S2 (breaching in the central part) were equal. , In scenario S1 the damage was higher corresponding to flood depth 0.1m to 0.5 m than to 0.5 to 1 m whereas in scenario S2 the
damage was higher corresponding to flood depth 0.5m to 1.0m than to 0.1 to 0.5m. From these findings it can be concluded that the flood extent, flood depth and damage depended on the breach location. Moreover, the comparison of the flood damages and flood extent led to identification of scenario S1 as the worst case scenario and the western part of the sea facing dike as the critical location for breaching.

The scenarios considering the effect of climate change (sea level rise) indicated that the flood extent and damage due to flood
will increase with sea level rise.

Flood risk was calculated as the product of occurrence probability of flood event and negative consequences (damage). The generated flood risk maps indicated that for all the scenarios areas adjacent to the dike and canals inside the polder had higher

risk of flooding. For better access to the canals, for transportation and livelihood, development of infrastructure and households nearby the canals increases the vulnerability. Similarly, developing land for infrastructure and household on the country side of the dikes increases vulnerability. Combining effect of increased vulnerability and higher flood depth results in elevated risk of flooding due to dike breach during a cyclone.

Inundation maps of all 72 scenarios were compared to generate the probabilistic flood map, which indicated that the areas with rice fields are the least and the settlements are the most probable areas to be flooded. Although the inhabitants are mostly dependent on agriculture, the flooding of settlements will cause most damage and force relocation.

Measured storm surge level for previous cyclones were unavailable. Therefore, for this research synthetic water level time series was generated considering the storm surge height presented by Islam *et al.* (2013), for a cyclone of 25 year return period.

The probability of flooding in a protected area is complicated and it was assumed that the probability of storm surge occurrence and breaching are the same. Limited number of field observations were available to compare the results of 2D model. The limited calibration possibility of hydraulic models stresses the importance of field observations pre-, post- and during-flood events. As the future land use data were not available, current land use has been used for the future scenarios as well.

The primary objective of the research was to present a methodology for generating FRM and PFM for the breaching of dike
during a cyclone. Due to the lack of data on the existing condition and previous history of breaching of the dike, the probability of the dike breaching could not be determined. Comprehensive surveys should be conducted to determine the physical condition of the existing embankments and their breach history. Using this data, a joint probability of flooding due to storm surges and breaching may be considered in future studies. As the sea beach outside the dike on the sea side was not included in the 2D model, the effect of mangrove forest could not be determined. A single breach location was considered for all the
developed scenarios. The probability of multiple dike breaching for a polder should be studied as well. Moreover, due to lack of data, the storm surge height for the present scenarios was used for future scenarios as well. As the sea surface temperature will change in the future due to climate change, the height and intensity of the storm surges will be affected as well. Research on the change of storm surge height and intensity due to climate change should be conducted in the future. Bathymetric data with coarse grid resolution from GEBCO was used as the measured bathymetric data for the sea was not available. Furthermore,
the study relied on the previous literatures for developing depth damage curves. Conducting field survey to generate these curves will provide more reliable damages due to flood. The developed and simulated model depended on the field measurements and logical assumptions which might be the source of errors. For damages, only direct damages were included. Inclusion of indirect damages will provide more realistic estimates.

Bangladesh is a hazard prone country and cyclone induced storm surge is one of many natural disasters experienced by the
coast of Bangladesh. The storm surges cause severe damage to the earthen embankment/dikes protecting the coastal polders. The methodology presented in this paper to develop the 1D-2D inundation model, PFM, risk maps and to identify the critical locations for breaching can assist in better preparedness against flooding and help in damage reduction by land use zoning and management. At present, the PFM and FRM due to storm surges and breaching of the dikes are not available for the coastal polders.

Climate change will likely cause increase in the frequency and intensity of the cyclone around the world. This will call for large investments for the improvement of existing or new protection works for the deltas. Identification and prioritising maintenance of critical locations of dike breaching can potentially prevent a disaster. Non-structural tools such as land use zoning with the help of flood risk maps and probabilistic flood maps have the potential to reduce the risk and the damage due to dike breaching. The method presented in this research can potentially be utilized for the deltas around the world to reduce vulnerability and flood risk due to dike breach caused by cyclone induced storm surges.

## 6. Acknowledgements

The support of Institute of Water Modelling (IWM), Dhaka, Bangladesh, in providing the surveyed data is gratefully acknowledged.

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
