# Peer review of "Flood risk assessment due to cyclone induced dike breaching on coastal areas of Bangladesh"

_Natural Hazards and Earth System Sciences, 2018_

## Referee Comment (RC1) · Anonymous Referee #1 · 14 Jul 2018

**Review of manuscript:** "*Flood risk assessment due to cyclone induced dike breaching on coastal areas of Bangladesh*".

**Overview:**

The paper presents a method for Probabilistic Flood Mapping in a very interesting case study area, finding areas of high-risk within the polder, based on developed scenarios. Some of the methods to reach these results are somewhat simplified, but do give a good overview to the single-case study, and have the potential to be applied in other locations.

However, I find the conclusions out of scope, and not in-line with what has been done in the study. Multiple grammatical and formatting errors, ambiguous figures, and a very unclear methodology and message all make the manuscript difficult to read, and major revisions are suggested before this work can even be properly assessed.

A comprehensive rewrite by the authors would allow for a more thorough review, and would benefit the article itself. It should be noted however, that enhanced clarity in the methods used are likely to raise more questions from a reviewer.

**General Comments:**

The main issues I have with the paper are as follows;

- Methodology: The methods used for calibration, breaching analysis and scenario development are all unclear and open to debate, but the biggest problem is the 1D2D model used. It is not described clearly, and as I understand it, models the sea in the 1D component. If this is the case, it requires a much better explanation and/or figures.
- Message: I find the message of the paper ambiguous. The discussion and start of the conclusion mention the dynamics of the case-study, which make sense as discussion topics. However, conclusions about flood forecasting and early warning systems seem out of place. Perhaps the potential development of PFMs for other polders is better suited to be discussed in the conclusions. Lines such as 'end the problem of poverty' should certainly be reconsidered.
- English: Multiple mistakes are found which distract the reader and give the impression of a careless approach to the work. In the specific comments below, only the ones found in the introduction are listed, but many more exist throughout the manuscript.

**Specific Comments:**

**Abstract**

- P1, Line 22: Presumably this second smaller abstract is not meant to be part of the main abstract. This is perhaps a formatting error during the upload process.

**Introduction**

- P2, Line 7: "…to protect the land from flooding due to diurnal high tide". The English here is incorrect. Either 'high tides' or 'the diurnal high tide'. Also, do the polders not also protect from the heavy rainfall mentioned just before this sentence?
- P2, Line 14: "Rising the crest level...". English.
- P2, Line 16: "Effect of…". English
- P2, Line 16: "Moreover, non-structural flood mitigation measures such as (…) and (…) is currently unavailable for the coastal areas of Bangladesh". English.
- P2, Line 23: "Furthermore..". This sentence suggests that SLR is not an effect of climate change. Did Mendelsohn et al. include this in their study?
- P2, Line 31: I cannot find where the variables of breach width, height and propagation are analysed in the study
- P2, Line 31: Scenarios mentioned only previously in abstract. Authors could consider a minor revision here

**Study Area**

- P3, Fig. 1: Upazilla term used in figure, but not explained in text. Presumably it is a form of district
- P3, Fig. 1: Would it be possible to indicate the extent of the mangrove forests?
- P3, Line 7: Who has classified this? The authors or a governmental body?

**Methodology**

- P4, Line 18: "…simulated using discharge as the upstream boundary…". What discharge? Is it important? Is it correlated to the cyclonic rainfall? Is it negligible in relation to the water level.
- P4, Line 29-30: "…and the location furthest from the dike breach is most sensitive." Given we don't (yet) know the locations of the breach or in which direction from the breach you mean, this is very ambiguous. You presumably mean in areas of low flow.
- Side Note: I think the numbers should continue throughout the article, not restart after every page. Perhaps this is an article format, if so, please ignore this comment.
- P5, Line 3: This paragraph about data gathering seems out of place, considering that data gathering was described before the previous paragraph about sensitivity analysis.
- P5, Line 6: Perhaps you should mention that the flood extent data from MODIS data was (presumably) used for calibration.
- P5, Fig. 2. Please indicate the Khaprabhanga river on the map

- P5, Fig. 2. As I understand it, the 1D component of the model stretches right around the polder, from the start of the Khaprabhangra river into the foreshore. Can you indicate the extents on the map?
- P5, Line 12: "For the rivers, the surveyed cross…". You are presumably referring to the Andharmanik and Galachipa rivers on the east and west sides of the polders, but the previous sentence mentions only river. Please clarify this.
- P5, Line 13: I find the use of 1D channels to simulate the foreshore very irregular, and feel it deserves more explanation or references of previous methods. Are these channels connected to the river channels? Is discharge a factor? It is not mentioned
- P5, Line 14: 13 control structures are mentioned here, which are presumably the 'Sluice gates' indicated on the map. If they are, please use the same term, and also, why are 13 not indicated?
- P5, Line 16: "Therefore, the canal network inside the polder was not included in the 1D model". The canals will have no effect on the dynamics outside the polder, but once flooding occurs they almost certainly will. I understand the DEM resolution will be too coarse to capture them, but this fact should be mentioned.
- P5, Line 19: Surely the foreshore data has no average slope?
- P6, Line 13: Are these storm surge heights directly applied as boundary conditions to the 1D model at every 1D cross-section location on the foreshore. I find this very difficult to understand.
- P6, Line 16: This seems out of place, perhaps more suited to the literature review earlier.
- P6, Line 26: Where is this section? As mentioned it should be in the map
- P6, Line 30: It was previously indicated that the breach geometry and propagation were variable in the scenario make-up (Abstract and Introduction). However in the end they are dependent on the other variables. This should be made clear
- P7, Line 6: Why not call the scenarios east west and central for simplicity?
- P7, Table 1: The SLR variation is based on current conditions and a possible future rise in 2100. This raises the question as to which period the PFMs that have been developed correspond. Perhaps it makes more sense to vary SLR for a given future moment according to the RCP scenarios. Also, the 1/25yr surge height used for the cyclone is presumably for current conditions, but as you explain earlier, this is subject to change.
- P7, Line 9: I don't understand this. If flooding results from the 3 worst case scenarios are available, it surely means breach locations are already selected. So how does this allow for a critical breach location to be selected?  Is this flooding from overtopping of the dikes?
- P8, Line 3: "…depth-damage curves from elsewhere." This is explained later, but at this point the sentence is very ambiguous.
- P9, Line 5: "The critical location of breaching…" Why is this included here? It adds to my existing confusion about how these locations are selected.
- P10, Line 2: "we have assumed that the probability of occurrence of the hazard and the probability of failure of dike as the same". Can the authors estimate the accuracy of this assumption? Presumably no flooding occurs (from overflow) of the dike in the simulations without breaching, but perhaps wave overflow would occur?
- P10, Line 4: I don't understand the relevance of this reference, as all scenarios used in the study have the (assumed) same probability.

- P10, Line15: "…comparing the observed and simulated water level and discharge". As mentioned previously, you have not mentioned what discharges are being simulated, or what you are calibrating them to. Also if the cyclone water levels are applied as boundary conditions, surely the calibration is trivial?
- P14, Figure 7: Can the authors explain why the damage decreases for larger flood depths?
- P15, Line 6: Ignoring depths less than 0.5m seems quite extreme, can the authors explain why this was done?
- P16, Line 22: "Figure 4 demonstrates that the depth of flooding gradually decreases as the water moves inland". This is not true, the figure only shows inundation extent. Perhaps the authors mean imply.
- P17, Line 18: "…(Fig. 8). Therefore, although canals play a crucial role in the economy and social life of the area, they also increase the risk of flooding". This is a strange, and in my view, inaccurate conclusion. Figure 8 shows the residential areas as high risk because they are more valuable. They happen to be situated beside canals.

---

## Referee Comment (RC2) · Anonymous Referee #2 · 30 Jul 2018

**Review of manuscript:** *"Flood risk assessment due to cyclone induced dike breaching on coastal areas of Bangladesh".*

Overview

The paper describes the methods and suggests tools for the probabilistic flood mapping in a polder area of Bangladesh. The study area selected by the author is interesting in terms of its geographical complexity and challenges related to the data collection. The methods used are rather simplified and aimed at giving a general overlook on the problem.

The main concerns

There are however, some major concerns about the idea behind methods and scenarios selection. The research questions should be addressed in Discussion section. One of the main problems is the description and structure of Study area and Methods sections. Some additional references are required in places where it is not clear where exactly the data or information come from. In addition, there is a large amount mistakes in language usage, both grammar, punctuation and word selection. The figures are not consistent throughout the manuscript. Therefore, my recommendation is to return this manuscript to authors for major revisions

General comments

Study area. Due to the specific conditions of the region, it is important to give more clarity and structure to this section. Probably it is a good idea to consider removing some unnecessary information and add more visualisation to more important aspects that are crucial for this specific research.

Research question. Should be stated clearly what exactly is developed within the study and to which degree it is considered innovative.

Methodology. Here are major rewritings are required to increase the quality of the paper. Modelling sub-section needs more clarifying in tools selection and usage. In addition, I suggest more description of the data used for model set-ups and calibration.

The subsection 3.2 Cyclonic scenarios considered; the selection of the values for different scenarios based on the IPCC report is rather subjective. It is suggested to consider regional sea level changes rather than global mean, as there is a significant difference specifically for Bangladesh. This may bring more impact on the outcomes of the study.

Discussion and Conclusion. It would be worth writing how/if the future studies would improve the current outcomes.

The take-home message is rather vague. The discussion section needs major re-writing in accordance to the research questions stated in Introduction. In my opinion such general methods used in this study should be accompanied with rather more detailed (sub)-section on the sources of errors and limitations.

Heroic assumptions such as "lead to economic growth" and "end the problem of poverty" should be avoided.

English. A serious revision of the language is necessary to improve the quality and readability of the manuscript. Among main issues I would outline: plural vs. singular, passive voice use, punctuation, repetitions of the same structures in consecutive sentences/paragraphs, repetitions of abbreviation explanations, articles selection, language use, etc..  The specific remarks do not cover language issues.

Specific remarks

p.2 line 2. According to Neumann et al (2015) 49% of population located in low elevated coastal zone for the year 2000, at that time the overall population of Bangladesh was 139 mil. Values should be corrected.

p.2 line 11. The number US$1.67 million seems rather small, needs additional check.

p.2 line 14. "Raising the crest level …" the sentence is unclear.

p.2 line 15. References needed to indicate which exactly previous studies were done in this matter.

p.2 line 17. It needs more clarification how land use zoning address the flood mitigation.

p.2 line 19. "…of these tropical cyclones will increase…" the statement *will* is rather confident, however it is *likely* increase. We are not 100% sure it will increase the intensity of storms. Look further through the manuscript for same errors.

p.2 line 26. Which exactly severe consequences specifically in Bangladesh? Look at Neumann et al (2015) for ideas.

p.3 line 3. It is recommended to visualise coordinates in Figure 1.

p.3 line 6. The source of census data is missing.

p.3 line 12. Consider the importance of putting the local names of seasons to the manuscript.

p.3 line 15. Some figures on the land subsidence rates may bring more light on the severity of the problem in the region.

p.4 line 12. "Model set up" rather than "model development"

p.4 line 20. The reference on FINMAP is missing.

p.4 line 26. More details on the computation mesh are recommended.

p. 5 line 9. The version of the model is missing.

p.6 line 17. I would include the figures on the land subsidence.

p.6 line 23. The figures of SLR indicated could be updated to the ones for 2100.

p. 7 line 15. It is better not to describe indirect damages if they are not consider further.

p.8 line 16. More reasoning for choosing of the figure of 50% would bring more light on the selection.

p.9 line 7. "More research…" is rather suitable for conclusion.

p.9 line 16. This definition of risk was presented earlier by Helm 1996. See Helm, P. (1996). Integrated Risk Management for Natural and Technological Disasters. Tephra, 15(1), 4-13.

p.10 line 9. It is not clear where M and N are in your formula.

p.11 Figure 4. The boundaries are not clear, some simplification of shapes could bring more readability to the map.

p.12 Figure 5. There is some confusion what exactly this figure is supposed to show.

p.15 Figure 8. The map layout is not consistent with other maps.

p.15 line 6. Some elaborate clarification why 0.5 m is used. My guess, some damages might be underestimated by selecting such high value.

---

## Author Comment (AC1) · 30 Jul 2018

Authors would like to thank the reviewer for the time taken to review the paper and for the suggestions and comments made. These were found very helpful to improve the manuscript.

Please find bellow the answers to the comments.

Remark: Reviewer comments are normal text, authors' response is italic text.

**General Comments from the reviewer:**

The main issues I have with the paper are as follows;

- **Methodology:** The methods used for calibration, breaching analysis and scenario development are all unclear and open to debate, but the biggest problem is the 1D2D model used. It is not described clearly, and as I understand it, models the sea in the 1D component. If this is the case, it requires a much better explanation and/or figures.

*Author's Response:*

*Thank you for the comment. More detailed information will be provided in the methodology with a flow chart to bring more clarity in the revised version, following your suggestion.*

- **Message:** I find the message of the paper ambiguous. The discussion and start of the conclusion mention the dynamics of the case-study, which make sense as discussion topics. However, conclusions about flood forecasting and early warning systems seem out of place. Perhaps the potential development of PFMs for other polders is better suited to be discussed in the conclusions. Lines such as 'end the problem of poverty' should certainly be reconsidered.

*Author's Response:*

*More effort will be put in to create more coherent and connected conclusion in the revised version.*

- English: Multiple mistakes are found which distract the reader and give the impression of a careless approach to the work. In the specific comments below, only the ones found in the introduction are listed, but many more exist throughout the manuscript.

*Author's Response:*

*To improve the level of English, native English speakers will be consulted and the revised version of the manuscript will be adjusted following their suggestion.*

**Reviewer 1:**

**"**P1, Line 22: Presumably this second smaller abstract is not meant to be part of the main abstract. This is perhaps a formatting error during the upload process.**"**

*Authors' Response:*

*The error will be adjusted accordingly.*

**Reviewer 1:**

**"**P2, Line 7: "…to protect the land from flooding due to diurnal high tide". The English here is incorrect. Either 'high tides' or 'the diurnal high tide'.**"**

*Authors' Response:*

*Thank you for the suggested correction. To improve the level of English, native English speakers will be consulted and the new version of the manuscript will be adjusted following their suggestion.*

**Reviewer 1:**
"Also, do the polders not also protect from the heavy rainfall mentioned just before this sentence?**"**

*Authors' Response:*

*Most of the polders of Bangladesh were built under the project titled as Coastal Embankment Project (CEP) in the 1960s. However, several articles and reports such as Mondal et al., 2006, Islam 2006, Islam et al., 2013, Bangladesh Delta Plan 2100, Coastal Embankment Improvement Project (Phase I) (Main Report) etc. indicated that the polders were constructed to protect the land from tidal flooding and salinity intrusion.*

**Reviewer 1:**
"P2, Line 14: "Rising the crest level...". English.**"**

*Authors' Response:*

*As mentioned previously English will be specially checked for improvement.*

**Reviewer 1:**
"P2, Line 16: "Effect of…". English**"**

*Authors' Response:*

*English will be checked and corrected as needed.*

**Reviewer 1:**
"P2, Line 16: "Moreover, non-structural flood mitigation measures such as (…) and (…) is currently unavailable for the coastal areas of Bangladesh". English.**"**

*Authors' Response:*

*English will be checked and corrected as needed.*

**Reviewer 1:**
"P2, Line 23: "Furthermore..". This sentence suggests that SLR is not an effect of climate change. Did Mendelsohn et al. include this in their study?**"**

*Authors' Response:*

*Authors acknowledge that the Sea Level Rise is indeed result of climate change and the paragraph will be rearranged to clarify the message. Moreover, English will be checked and corrected as needed.*

**Reviewer 1:**
"P2, Line 31: I cannot find where the variables of breach width, height and propagation are analysed in the study**"**

*Authors' Response:*

*As the reviewer indicated in the later comments that the breach properties were not independent variables, it should not be included as a parameter of scenario development. Authors acknowledge the suggested correction and the revised version will be adjusted accordingly.*

**Reviewer 1:**
"P2, Line 31: Scenarios mentioned only previously in abstract. Authors could consider a minor revision here**"**

*Authors' Response:*

*Authors acknowledge the suggestion and will be adjusted accordingly.*

**Reviewer 1:**
"P3, Fig. 1: Upazilla term used in figure, but not explained in text. Presumably it is a form of district**"**

*Authors' Response:*

*Thank you for raising this issue the manuscript will be adjusted accordingly.*

**Reviewer 1:**
"P3, Fig. 1: Would it be possible to indicate the extent of the mangrove forests?**"**

*Authors' Response:*

*The extent of the mangrove forest will be indicated in the map following your suggestion in the revised version.*

**Reviewer 1:**

"P3, Line 7: Who has classified this? The authors or a governmental body?**"**

*Authors' Response:*

*The classification was done by the Ministry of Land of Bangladesh. The information will be added in the revised version.*

**Reviewer 1:**

"P4, Line 18: "…simulated using discharge as the upstream boundary…". What discharge? Is it important? Is it correlated to the cyclonic rainfall? Is it negligible in relation to the water level.**"**

*Authors' Response:*

*The authors were trying to describe the developed 1D model which was calibrated against the measured water level and discharge on the river stations and then couples with 2D model. The 1D model was calibrated for normal condition (without cyclones) as hydrometric data during cyclonic events were not available. Moreover, the simulated cyclonic event "Sidr" made landfall in the coast of Bangladesh during the month of November which is post monsoon season. In the past, most of the cyclones his the coast of Bangladesh during the months of October-November or April-May which are pre and post monsoon where discharge from rivers don't play a significant role. The authors acknowledge the "Methodology" section requires more detailed and clearer description and will try to do so in the revised version.*

**Reviewer 1:**

"P4, Line 29-30: "…and the location furthest from the dike breach is most sensitive." Given we don't (yet) know the locations of the breach or in which direction from the breach you mean, this is very ambiguous. You presumably mean in areas of low flow.**"**

*Authors' Response:*

*As suggested by the reviewer, authors intention was to indicate about the sensitivity of the low flow areas to the coefficient of roughness "n", indeed.*

**Reviewer 1:**

"P5, Line 3: This paragraph about data gathering seems out of place, considering that data gathering was described before the previous paragraph about sensitivity analysis.**"**

*Authors' Response:*

*Authors will adjust accordingly in the reviewed version.*

**Reviewer 1:**

"P5, Line 6: Perhaps you should mention that the flood extent data from MODIS data was (presumably) used for calibration.**"**

*Authors' Response:*

*It will be adjusted accordingly.*

**Reviewer 1:**
"P5, Fig. 2. Please indicate the Khaprabhanga river on the map**"**

*Authors' Response:*

*It will be adjusted accordingly.*

**Reviewer 1:**
"P5, Fig. 2. As I understand it, the 1D component of the model stretches right around the polder, from the start of the Khaprabhangra river into the foreshore. Can you indicate the extents on the map?**"**

*Authors' Response:*

*It will be adjusted accordingly.*

**Reviewer 1:**
"P5, Line 12: "For the rivers, the surveyed cross…". You are presumably referring to the Andharmanik and Galachipa rivers on the east and west sides of the polders, but the previous sentence mentions only river. Please clarify this.**"**

*Authors' Response:*

*In the new version of the manuscript it will be clarified and adjusted accordingly.*

**Reviewer 1:**
"P5, Line 13: I find the use of 1D channels to simulate the foreshore very irregular, and feel it deserves more explanation or references of previous methods. Are these channels connected to the river channels? Is discharge a factor? It is not mentioned."

*Authors' Response:*

*Thank you for the comment. The authors tried to represent the condition of the water bodies adjacent to the dikes with 1D model with synthetic boundary condition. As the coast of Bangladesh is flat and shallow, a 2D model for coastal hydrodynamics will require inclusion of larger area of the sea which was not the area of interest and which would have increased the simulation time too. The utilised 1D network was not connected with the rivers as explained earlier the river water didn't play a major role during the previous cyclones as it's pre or post monsoon. The authors acknowledge this requires clearer description. The revised version will be adjusted to provide more detailed and clearer explanation.*

**Reviewer 1:**

"P5, Line 14: 13 control structures are mentioned here, which are presumably the 'Sluice gates' indicated on the map. If they are, please use the same term, and also, why are 13 not indicated?**"**

*Authors' Response:*

*It will be corrected accordingly.*

**Reviewer 1:**

"P5, Line 16: "Therefore, the canal network inside the polder was not included in the 1D model". The canals will have no effect on the dynamics outside the polder, but once flooding occurs they almost certainly will. I understand the DEM resolution will be too coarse to capture them, but this fact should be mentioned.**"**

*Authors' Response:*

*The canal network inside the polder was not connected to the river network as during a cyclone the gates of the control structure will remain closed. But the larger canals were included in the DEM. The width of the larger canals were wide enough to be included in the DEM.*

**Reviewer 1:**

"P5, Line 19: Surely the foreshore data has no average slope?**"**

*Authors' Response:*

*The authors agree that the rivers had higher slope than the foreshore area. As the distance between computational points is inversely proportional to slope, the rivers will require smaller Δx and same Δx will reduce instability for the foreshore area too. A clearer explanation will be provided in the revised version.*

**Reviewer 1:**

"P6, Line 13: Are these storm surge heights directly applied as boundary conditions to the 1D model at every 1D cross-section location on the foreshore. I find this very difficult to understand.**"**

*Authors' Response:*

*To ensure same water level at all the points of the foreshore reach, same water level was applied at both ends of the reach as hydro dynamic boundaries. The authors acknowledge this requires clearer description. The revised version will be adjusted to provide more detailed and clearer explanation.*

**Reviewer 1:**

"P6, Line 16: This seems out of place, perhaps more suited to the literature review earlier."

*Authors' Response:*

*Thank you for the suggestion. Agreed that part of the paragraph can also be suited in the literature review section and will be adjusted accordingly.*

**Reviewer 1:**
"P6, Line 26: Where is this section? As mentioned it should be in the map"

*Authors' Response:*

*Following your suggestion, it will be depicted in the map with mangrove forest,.*

**Reviewer 1:**
"P6, Line 30: It was previously indicated that the breach geometry and propagation were variable in the scenario make-up (Abstract and Introduction). However in the end they are dependent on the other variables. This should be made clear"

*Authors' Response:*

*Indeed these are not independent variables and should not be stated as parameters for scenario development. The revised version will be adjusted accordingly.*

**Reviewer 1:**
"P7, Line 6: Why not call the scenarios east west and central for simplicity?"

*Authors' Response:*

*The change of the tittle of the scenarios will require the adjustment of most of the figures and tables. Because this is just a matter of choice, authors will see what can be done in the new version of the manuscript.*

**Reviewer 1:**
"P7, Table 1: The SLR variation is based on current conditions and a possible future rise in 2100. This raises the question as to which period the PFMs that have been developed correspond. Perhaps it makes more sense to vary SLR for a given future moment according to the RCP scenarios. Also, the 1/25yr surge height used for the cyclone is presumably for current conditions, but as you explain earlier, this is subject to change."

*Authors' Response:*

*Thank you for your suggestion. As stated earlier SLR has been considered according to the RCP scenarios. The storm surge height for 1/25 yr event was based on current condition. As not enough measured data was available, previous literatures were used for determining the storm surge height*

*for simulation. Time series data for sea level rise for the study area was not available either. Therefore, the scenarios were developed for the years for which data was available. The authors acknowledge that the section needs to be described more clearly and will be done in the revised version.*

**Reviewer 1:**
"P7, Line 9: I don't understand this. If flooding results from the 3 worst case scenarios are available, it surely means breach locations are already selected. So how does this allow for a critical breach location to be selected? Is this flooding from overtopping of the dikes?"

*Authors' Response:*

*The flooding occurred for breaching only as the crest level of the dike was considered to be elevated to a height suggested by CEIP in the future. Breaching of the dike was considered for all the scenarios, therefore, authors agree that dike was breached already in the three worst case scenarios as well. The intention was to present a methodology to compare and identify a critical location based on damaged caused by flood in case of breaching of dike which could be applied in other locations. Providing better protection at the critical locations might reduce the damage significantly during and after a cyclone. The authors will provide clearer explanation in the revised version.*

**Reviewer 1:**
"P8, Line 3: "…depth-damage curves from elsewhere." This is explained later, but at this point the sentence is very ambiguous."

*Authors' Response:*

*It  will be adjusted accordingly.*

**Reviewer 1:**
"P9, Line 5: "The critical location of breaching…" Why is this included here? It adds to my existing confusion about how these locations are selected."

*Authors' Response:*

*It will be adjusted accordingly. The locations were selected on the basis of exposure to the storm surge and placement along the sea facing dike (east, west and central). Will be explained in better detailed in the revised version.*

**Reviewer 1:**
"P10, Line 2: "we have assumed that the probability of occurrence of the hazard and the probability of failure of dike as the same". Can the authors estimate the accuracy of this assumption? Presumably no flooding occurs (from overflow) of the dike in the simulations without breaching, but perhaps wave overflow would occur?"

*Authors' Response:*

*The crest level of the dikes was considered to be at the design height suggested by Islam et al., 2013 and CEIP. Wave action was considered during the calculation of design crest level and a free board was also considered. It can be safely concluded that the crest level suggested by Islam et al., 2013 will be sufficient enough to protect area inside the dike from overtopping during a 1/25 year event. Moreover, the breaching of a dike depends on the physical condition of the dikes and it's soil properties. Neither of these data were available. Therefore, the calculation of probability of the dike breach was not possible. To simplify and to investigate the effect of dike breach, the breaching probability was considered same as the cyclonic event. This part will be described in more detailed in the revised version.*

**Reviewer 1:**
"P10, Line 4: I don't understand the relevance of this reference, as all scenarios used in the study have the (assumed) same probability"

*Authors' Response:*

*The authors intention was to provide reference for the equation used for calculation of probabilistic flood maps. It will be rephrased in the new version of the manuscript, for better clarity.*

**Reviewer 1:**
"P10, Line15: "…comparing the observed and simulated water level and discharge". As mentioned previously, you have not mentioned what discharges are being simulated, or what you are calibrating them to. Also if the cyclone water levels are applied as boundary conditions, surely the calibration is trivial?"

*Authors' Response*

*As mentioned earlier the calibration of 1D model was done for normal condition, not for a cyclone event as no data was available during that event and very few data was available for calibrating the 2D model. The intention of the authors was to present a methodology with which flood risk maps and PFMs can be generated for different locations. This section will be described in more detailed in the revised version.*

**Reviewer 1:**
"P14, Figure 7: Can the authors explain why the damage decreases for larger flood depths?"

*Authors' Response*

*The damage is a function of flood depth but the unit is per unit area(per $m^2$). Therefore, if the flood extent for higher depth is lower, the damage due to flood might be lower too.*

**Reviewer 1:**

"P15, Line 6: Ignoring depths less than 0.5m seems quite extreme, can the authors explain why this was done?"

*Authors' Response*

*The developed damage curves suggest that the damage for flood depth below 0.5 m is minimal. Moreover, the authors tried to explore the effect of living with flood concept. Also tried to consider the uncertainty of the DEM and the 2D inundation model. The depth as 0.5m is an arbitrary depth. For a country where flood is a recurrent phenomenon with larger depth, living with flood might be already adopted by the local people.*

**Reviewer 1:**

"P16, Line 22: "Figure 4 demonstrates that the depth of flooding gradually decreases as the water moves inland". This is not true, the figure only shows inundation extent. Perhaps the authors mean imply."

*Authors' Response*

*It will be adjusted accordingly.*

**Reviewer 1:**

"P17, Line 18: "…(Fig. 8). Therefore, although canals play a crucial role in the economy and social life of the area, they also increase the risk of flooding". This is a strange, and in my view, inaccurate conclusion. Figure 8 shows the residential areas as high risk because they are more valuable. They happen to be situated beside canals."

*Authors' Response*

*Indeed the residential areas had higher depth damage ratio than other land classes. But the residential areas were flooded primarily for being by the side of the canal. The authors' intention was to state that as these areas are adjacent to the canals for various reasons such as being advantageous for transportation, also makes them susceptible to flooding and higher damage. The authors will try to explain more clearly in the revised version.*

**Reference**

Mondal, M.K., Tuong, T.P., Ritu, S.P., Choudhury, M.H.K., Chasi, A.M., Majumder, P.K., Islam, M.M. and Adhikary, S.K., 2006. Coastal water resource use for higher productivity: participatory research for increasing cropping intensity in Bangladesh. *Environment and Livelihoods in Tropical Coastal Zones: Managing Agriculture-Fishery-Aquaculture Conflicts.*, pp.72-84.

Islam, M.R., 2006. 18 Managing Diverse Land Uses in Coastal Bangladesh: Institutional Approaches. *Environment and livelihoods in tropical coastal zones*, p.237.

Islam, M.S., Alam, R., Khan, M.Z.H., Khan, M.N.A.A. and Jahan, S.N., 2013. Methodology of crest level design of coastal polders in Bangladesh. In *4th International Conference on Water & Flood Management*.

---

## Author Comment (AC2) · 27 Aug 2018

Authors would like to thank the reviewer for the time taken to review the paper and for the suggestions and comments made. These were found very helpful to improve the manuscript.

Please find bellow the answers to the comments.

Remark: Reviewer comments are normal text, authors' response is italic text

**Review of manuscript:** *"*Flood risk assessment due to cyclone induced dike breaching on coastal areas of Bangladesh".

**Overview**
The paper describes the methods and suggests tools for the probabilistic flood mapping in a polder area of Bangladesh. The study area selected by the author is interesting in terms of its geographical complexity and challenges related to the data collection. The methods used are rather simplified and aimed at giving a general overlook on the problem.

*Authors: Thanks for the comments*

**The main concerns**

There are however, some major concerns about the idea behind methods and scenarios selection. The research questions should be addressed in Discussion section. One of the main problems is the description and structure of Study area and Methods sections. Some additional references are required in places where it is not clear where exactly the data or information come from. In addition, there is a large amount mistakes in language usage, both grammar, punctuation and word selection. The figures are not consistent throughout the manuscript. Therefore, my recommendation is to return this manuscript to authors for major revisions.

*Authors: We acknowledge the concerns raised by the reviewer. We will revise the manuscript to bring more clarity in the description of the study area and the presented methodology. We will further update the discussion section with the research questions. More specific answers are provided in the following sections.*

**General comments**

Reviewer:

Study area. Due to the specific conditions of the region, it is important to give more clarity and structure to this section. Probably it is a good idea to consider removing some unnecessary information and add more visualisation to more important aspects that are crucial for this specific research.

*Authors: The authors acknowledge the suggestion. More specific information about the study area will be provided and unnecessary information will be removed to improve the description of the study area.*

Reviewer:

Research question. Should be stated clearly what exactly is developed within the study and to which degree it is considered innovative.

*Authors: The authors understand that the innovation remained unclear in the draft. The main innovation of the research will be presented clearly in the revised manuscript.*

Reviewer:

Methodology. Here are major rewritings are required to increase the quality of the paper. Modelling sub-section needs more clarifying in tools selection and usage. In addition, I suggest more description of the data used for model set-ups and calibration.

*Authors: The authors acknowledge the suggestion. More detailed information will be provided in the methodology with a flow chart to bring more clarity in the revised version. The description of the data used for the setting up of the model and its calibration will be elaborated.*

Reviewer:

The subsection 3.2 Cyclonic scenarios considered; the selection of the values for different scenarios based on the IPCC report is rather subjective. It is suggested to consider regional sea level changes rather than global mean, as there is a significant difference specifically for Bangladesh. This may bring more impact on the outcomes of the study.

*Authors: The authors' intention was to present a global methodology applicable to everywhere. The relative sea level rise for RCP 8.5 of IPCC AR5 for the coast of Bangladesh is 0.56 m (GERIC, 2015) which is slightly lower than what was considered by the authors (0.63 m).*

Reviewer:

Discussion and Conclusion. It would be worth writing how/if the future studies would improve the current outcomes.

*Authors: The authors acknowledge the suggestion. The revised manuscript will have recommendation for future studies.*

Reviewer:

The take-home message is rather vague. The discussion section needs major re-writing in accordance to the research questions stated in Introduction. In my opinion such general methods used in this study should be accompanied with rather more detailed (sub)-section on the sources of errors and limitations.

*Authors: The discussion section will be adjusted with a reflection on the research questions. Limitations will be added in the research discussion section.*

Reviewer:

Heroic assumptions such as "lead to economic growth" and "end the problem of poverty" should be avoided.

*Authors: The authors acknowledge the suggestion and the phrases will be removed.*

Reviewer:

English. A serious revision of the language is necessary to improve the quality and readability of the manuscript. Among main issues I would outline: plural vs. singular, passive voice use, punctuation, repetitions of the same structures in consecutive sentences/paragraphs, repetitions of abbreviation explanations, articles selection, language use, etc.. The specific remarks do not cover language issues.

*Authors: To improve the level of English a native English speaker will be consulted.*

**Specific remarks**

Reviewer:

p.2 line 2. According to Neumann et al (2015) 49% of population located in low elevated coastal zone for the year 2000, at that time the overall population of Bangladesh was 139 mil. Values should be corrected.

*Authors: The authors acknowledge the suggestion and the manuscript will be adjusted accordingly.*

Reviewer:

p.2 line 11. The number US$1.67 million seems rather small, needs additional check.

*Authors: The authors thank the reviewer for pointing it out. The revised manuscript will be corrected accordingly.*

Reviewer:

p.2 line 14. "Raising the crest level …" the sentence is unclear.

*Authors: We agree and the revised line will be: "Raising the crest level was considered as the only mitigating measure".*

Reviewer:

p.2 line 15. References needed to indicate which exactly previous studies were done in this matter.

*Authors: The authors acknowledge the suggestion and the revised manuscript will be adjusted accordingly.*

Reviewer:

p.2 line 17. It needs more clarification how land use zoning address the flood mitigation.

*Authors: The authors think that land use zoning is widely used as a flood risk mitigation measure. We will add references to it.*

Reviewer:

p.2 line 19. "…of these tropical cyclones will increase…" the statement will is rather confident, however it is likely increase. We are not 100% sure it will increase the intensity of storms. Look further through the manuscript for same errors.

*Authors: Indeed, we will rewrite it as follows: "Moreover, the intensity and frequency of these tropical cyclones are likely to increase in the future due to climate change causing more damages."*

Reviewer:

p.2 line 26. Which exactly severe consequences specifically in Bangladesh? Look at Neumann et al (2015) for ideas.

*Authors: As suggested we will refer to Neumann et al. (2015).*

Reviewer:

p.3 line 3. It is recommended to visualise coordinates in Figure 1.

*Authors: We will add the coordinates in Figure 1.*

Reviewer:

p.3 line 6. The source of census data is missing.

*Authors: We thank the reviewer for pointing it out. The revised manuscript will mention the source of the census data.*

Reviewer:

p.3 line 12. Consider the importance of putting the local names of seasons to the manuscript.

*Authors: The local names of the copping seasons have been mentioned. However, the calendar months are mentioned as well and therefore, Authors think that any lack of clarity is not obvious.*

Reviewer:

p.3 line 15. Some figures on the land subsidence rates may bring more light on the severity of the problem in the region.

*Authors: We will search for any studies describing land subsidence rates and if available, we will report it.*

Reviewer:

p.4 line 12. "Model set up" rather than "model development"

*Authors: Thank you for the suggestions. As we are unable to identify the mistake, we will consult a native English speaker for clarity.*

Reviewer:

p.4 line 20. The reference on FINMAP is missing.

*Authors: We will add a reference to FINMAP.*

Reviewer:

p.4 line 26. More details on the computation mesh are recommended.

*Authors: The spatial resolution of the computational mesh has already been mentioned. It is unclear what additional details on the computational mesh are required. We will explore this issue.*

Reviewer:

p. 5 line 9. The version of the model is missing.

*Authors: We have used version 5 of the HEC-RAS tool. This information will be added in the revised manuscript.*

Reviewer:

p.6 line 17. I would include the figures on the land subsidence.

*Authors: As mentioned above, we will search for any studies describing land subsidence rates and if available, we will report it.*

Reviewer:

p.6 line 23. The figures of SLR indicated could be updated to the ones for 2100.

*Authors: The Authors' intention was to state the probable sea level rise suggested by IPCC. The authors will take a look and adjust the manuscript if required.*

Reviewer:

p. 7 line 15. It is better not to describe indirect damages if they are not consider further.

*Authors: We have not elaborated on indirect damages. We have just defined it to bring clarity.*

Reviewer:

p.8 line 16. More reasoning for choosing of the figure of 50% would bring more light on the selection.

*Authors: Muktadir and Hasan, 1985 stated in their study that the rural house hold of Bangladesh usually are built around a large country yard. The land classification by the Ministry of Land defines the whole house hold including the country yard as residential area. But the damage curve considered for the residential area was for the damage to the house and the country yard will not have significant damage if flooded. Considering this the authors tried to exclude the country yard from the residential area. The satellite image of the area indicated that the about half of the area of residential complex (house hold) is usually empty. Therefore, the figure 50% was considered.*

Reviewer:

p.9 line 7. "More research…" is rather suitable for conclusion.

*Authors: Authors acknowledge the suggestion and the manuscript will be adjusted accordingly.*

Reviewer:

p.9 line 16. This definition of risk was presented earlier by Helm 1996. See Helm, P. (1996). Integrated Risk Management for Natural and Technological Disasters. Tephra, 15(1), 4-13.

*Authors: Authors acknowledge the suggestion and the manuscript will be adjusted accordingly.*

Reviewer:

p.10 line 9. It is not clear where M and N are in your formula.

*Authors: The parameters M and N were defined in the manuscript following the equation. We will explore to bring further clarity.*

Reviewer:

p.11 Figure 4. The boundaries are not clear, some simplification of shapes could bring more readability to the map.

*Authors: Authors acknowledge the suggestion and will put more effort to increase the readability of the map.*

Reviewer:

p.12 Figure 5. There is some confusion what exactly this figure is supposed to show.

*Authors:*

*Authors' intention was to depict the effect of different variables of the scenario development. For example: to depict the effect of sea level rise, two scenarios with keeping all the other variables (except from sea level rise) constant and changing the SLR, were compared. Authors acknowledge that this requires clearer explanation and the revised manuscript will be adjusted.*

Reviewer:

p.15 Figure 8. The map layout is not consistent with other maps.

*Authors: The difference with other figures mostly appear due to the presence of legends in Figure 8. The purpose of this figure is also different from other figures and as such some dissimilarities in layout may not matter. We will look into the possibilities of bringing a consistent layout.*

Reviewer:

p.15 line 6. Some elaborate clarification why 0.5 m is used. My guess, some damages might be underestimated by selecting such high value.

*Authors: The developed damage curves suggest that the damage for flood depth below 0.5 m is minimal. Moreover, the authors tried to explore the effect of living with flood concept. Also tried to consider the uncertainty of the DEM and the 2D inundation model. The depth as 0.5m is an arbitrary depth. For a country where flood is a recurrent phenomenon with larger depth, living with flood might be already adopted by the local people. Moreover, this arbitrary was used for PMFs and the estimation of damage due to flood was not affected.*

**Reference**

Climate Service Center Germany (GERIC) and KfW Development Bank, 2015, CLimate Focus Paper: Regional Sea Level Rise South Asia, Hamburg and Frankfurt am Main.

Muktadir, M.A. and Hasan, D.M., 1985, December. Traditional house form in rural Bangladesh: a case study for regionalism in architecture. In Regional seminar on Architecture and the Role of Architects in Southern Asia (pp. 19-23).

---

## Referee Report (RR1)

**Review of manuscript:** "*Flood risk assessment due to cyclone induced dike breaching on coastal areas of Bangladesh*".

**Overview:**

The paper has improved significantly in all of the three main points mentioned previously (message, methodology and English). This, along with the improved figures, gives the reader a much better understanding of the nature of the research. However, the problems with these areas have not been fully resolved. Despite the much improved manuscript, in my opinion these issues still require major revisions to the methodology and the message. However, methodological changes or extra data analyses are not necessarily needed to achieve a publishable article.

**General Comments:**

The improvements from the previous manuscript, as well as the continued (or new) problems I observe in the 3 areas mentioned above are described below;

Methodology:

The methodology is much clearer in the second manuscript, and helped by the flowchart. However, there are 3 aspects that need to be addressed.

The first is the differentiation between the 3 extreme scenarios the 72 scenarios. I previously suggested that a different name be given for these sets of scenarios, but this was not done, and it requires careful analysis for the reader to dissect this for themselves. The flowchart helps a little but it also causes more questions, which are listed in the specific comments below.

The second aspect is the fact that all the outputs relate to a 1:25yr event. Highlighting this more clearly would help the reader to understand the context of the research in terms of a wider flood risk analysis for the region, (see comments on message, below). A serious problem not addressed with the results of this calculation event is that the authors do not define for which temporal period risks are applicable. For example, if the results represent the current risk, then of the 72 simulations, those simulations with future SLR should not be included. For the risk in 2100 (the date at which the SLR value is predicted to happen), the simulations with current sea level should not be used. Furthermore, subsidence over time was accounted for to estimate the current topography, but not future topography. This is briefly mentioned in the conclusions, but not the significance for the results are not.

Finally the 1D modelling explanation needs to be improved, especially in terms of boundary conditions. Please see the specific comments below.

Message:

"The primary objective of the research was to present a methodology for generating FRM and PFM for the breaching of dike during a cyclone". This line from the conclusion should have similar versions in the abstract and introduction, and in the opinion of this reviewer, should shape the overall message of the paper. It is also closer to the title of the paper. Instead the abstract and introduction sections describe the paper as "an investigation of the inundation pattern in a

protected area", which is an interesting case-study, but not innovative or of interest to a general audience. For example, a line of logic for the introduction could be;

- Bangladesh is susceptible to cyclones;
- Damage estimates and flood maps, as well the identification of critical breaching locations, are useful for mitigating risk
- Quantifying this data requires taking account of SLR, cyclone frequency etc. and potential for breaching.
- A method to do so for the 1:25yr event (design criteria), for a particular polder, is given below.

Then, as well as the specific results, the applicability of the method to other areas and return period events could be estimated.

English:

This has improved significantly, but again, continued errors distract the reader, especially in the conclusion. The paper cannot be accepted with so many mistakes, and it is strongly suggested to ensure they are removed before resubmission.

**Specific Comments:**

**Abstract**

- P1, Line 14: "Scenarios were developed by considering… geometrical properties of the breach, breach propagation time…". This line directly conflicts with P8, Line12: "As the geometry of the breach is not independent, it was not considered as a parameter for scenario development". This was highlighted previously and needs to be resolved.

**Introduction**

- P2, Line 26: "Furthermore, by the year 2100 the annual estimated damage due to tropical cyclones may increase by US$53 billion" This is surely due to the reasons the authors mention previously, and therefore not 'furthermore'.
- P2, Line 26: "study area is not benefitted with…" This sounds strange.

**Study Area**

- P3: Lines 9-13: This part seems just like a list of data, and doesn't connect well, consider revising.
- P4, figure 1: Please plot the extent of the dike on this map or on figure 3, as you mention it is only on the seaward side. Including the chainage points given in Figure 6 would also be useful.

**Methodology**

- P5, figure 2: The arrow from HEC-RAS to 'Estimate damage' during a loop is confusing. The output of 72 simulations from the loop should be used.
- P5, figure 2: Perhaps the damage and maps sections could be highlighted as outputs.
- P6, line 5: I don't think the HEC-Ras description here needs to be so detailed. In fact this whole paragraph should be reviewed for structure and cohesion.
- P7, line 3-5: The English is poor here, and it is unclear. Are all cells rectangular? How can they have a single resolution or area? This is made more confusing by the grid representation in the figure.
- P7: Figure 3: Presumably Khaprabhanga is more of a channel than a river, as it seems to connect two parts of the Bay of Bengal. If this is the case it should be mentioned.
- P7: Figure 3: How are the upstream and downstream locations selected? Presumably it is because it is assumed a cyclone always come from the west and therefore imposes a water level downstream, but this is never stated.
  If I am correct in this assumption of a water level timeseries as the main driver in the model, why wasn't a QH relationship used at the other end? This raises a number of questions about the water level profile along the seafront.
  Why was the Khaprabhanga included at all? No breaches occurred at that end, and it is not connected to the other 1D part.
- P7, line 16: Mangrove ForestS.
- P7: Line 17: "The storm surges…" English

- P8: Line 8: "…therefore, the wider and larger canals were included in the DEM". This sounds like the canals were 'burnt' into the DEM, which I understand was not the case.
- P8: Line 8: "As the geometry of the breach is not independent, it was not considered as a parameter for scenario development". As mentioned above this is contradicts another statement. Also, the geometry is independent of the water levels, the failure is not.
- P9, figure 6: Why is the middle chainage higher than the others during the rising limb? This raises more questions about the boundary conditions.
- P10, Line 13: The 1:25yr design criteria should be mentioned in the abstract and introduction, as it dictates the method and outputs.
- P10, line 18: from which direction is the angle measured, i.e. does angle of landfall of 230 degress indicate a cyclone moving from South to North? Apologies if this is an obvious question, but I am not hugely familiar cyclones.
- P10, Line 34: "Single breach was considered for each scenario". English.
  Also, perhaps the authors can suggest the consequences of the assumption of 1 breach for a 1/25yr storm. Is 1 an over or under estimate? How many breaches have been recorded before?
- P11, line 8; English
- P11, line 10; It is important to clearly identify what the maps and outputs represent. For example the PFMS are probabilistic inundations maps for a single event (1/25yr) in which a breach occurs.
- P12, line 23; The fact that you do not account for duration or velocity of inundation should be mentioned, especially if highlighting their importance.
- P14, line 2. English
- P14, line 7; "Pfj is the probability of reaching a certain storm surge level in simulation number j". Is each Pfj will be equal to 1/25 here? As mentioned I think it is important to specify these results relate to the 1/25yr level.

**Results and Discussion**

- P15, line 6: "…for identifying 5 the critical location of breaching". This could be confused as a process to identify locations for the modelling process. Consider revising, and/or changing to 'regions'.
- P16, figure 9: Presumably these figures are averages of subsets of the 72 simulations, This should be indicated in the text.
- P16, line 3: Now the authors move back to the subset of 3 scenarios. I find this hard to follow, especially since the names 'scenarios 1,2, and 3' do not relate to any cardinal numbers, but to breach locations on the west, centre and east.
- P18, figure 11: This figure and figure 10 need clarification. Can I infer from them that the area in which the maximum flood depth was less than 0.1m in scenario 1 (western breach) had an inundation of just above 2km$^2$, which causes about 400,000 euros of damage?  I see this is later explained in the discussion, but the authors might consider a clearer explanation at this point too.
- P18, line 3: Section 4.5 describes how the western breach was selected as the critical location, but you should mention that result here, as the flood map is clearly dependent on that. Perhaps it should also be plotted in figure 12 as well.

- P19, line 8; "This arbitrary threshold…". It has just been explained that the threshold is not arbitrary.
- P19, line 8; "…and the estimation of damage due to flood was not affected". I don't understand this sentence at all.
- P20, figure 13: Again I think it should be mentioned that this is the 1/25yr PFM.
- P20, line2: "Three worst case scenarios (Scenario 1, 2 and 3)…" Again there is confusion about these scenarios. They are not the 3 worst case scenarios, they are the worst case scenarios in terms of boundary conditions for each of the 3 potential breach locations.
- P20, line 3: "demonstrate that large area is flooded…". English
- P20, line 4: The text references figure 9, which relates to all 72 simulations, as I understand. This is in the middle of text about the 3 extreme scenarios, and further highlights the issue already mentioned.
- P20, line 5: "More than 25% of the total area of Polder 48 was inundated for the three scenarios...". Please make this sentence clearer, I understand it to mean 'At least 25% of the total area of Poder 48 was inundated in all 3 extreme scenarios"
- P20, line 10. It is not quite correct to say the risk is X million euro. Firstly it needs to be defined as a value per year or per event. Secondly these values are only the extreme versions of the 1/25yr event defined.
- P21, line 8: English
- P21, line 10: "due to breaching  of  unprotected  location  of  the  dike". English
- P21, line12: "Scenario 1 had higher risk of flooding as the damage due to flooding was maximum for Scenario 1". This sounds strange, consider revising.

**Conclusion**

- P22, line 1: "investigate inundation pattern in a polder". English
- P22, line 2: "were compared using total flooded area and estimated damage". For which scenario set? 72 or 3?
- P22, line 5: "Flood risk map and probabilistic flood map…". English. These mistakes are very distracting.
- P22, line 8: "during the breaching of central part of the sea facing dike". It doesn't happen during the breaching.
- P23, line 8: "Most flooded area had flood depth of 0.1 to 0.5 m…". English.
- P22, line 20: "…are least and most probable land use respectively". This sounds strange.
- P22, line 24: This is the first mention of 'synthetic water level time series'. It is clear from the bottom of figure 6 that they are present, but not how they were generated.
- P22, line 27:" This stresses the importance of field observation pre, post and during event". I don' see what actually stresses this importance.
- P22, line 30: "Comprehensive survey should be conducted…". English.
- P22, line 33 "…a join probability" English
- P23, line 1: "Single breach was considered for all the developed Scenarios". English
- P23, line 2: "…should be studies as well". English
- P23, line 7: "Conducting field survey to generate these curves…" English

---

## Referee Report (RR2)

**Second review of manuscript:** "*Flood risk assessment due to cyclone induced dike breaching on coastal areas of Bangladesh*".

Overview and general remarks

The second version of the manuscript is better than the first one. The authors put obvious effort in editing the text after the referee´s comments and suggestions. There are significant improvements in the Methods and Discussion/Conclusions section in terms of the content required for the study and readability. The Figures and Tables are clear. Most of language mistakes are corrected. Nevertheless, there are some improvements should be done, mostly related to refining some description in the Methods and Conclusion sections, missing/wrong reference and language. Therefore, I suggest to return the manuscript for minor revisions.

Specific remarks

Figure 12 and 13 miss a scale bar.

p.2 line 10. Were *recently* designed.

p.2 line 20. The intensity of tropical cyclones is *likely* to increase.

p.3 line 7. The land use is mainly ….. this sentence needs to be rephrased.

p.5 line 8. …(a company from Finland). In my opinion this information is unnecessary.

p.6 line 1. As was pointed out in the previous round of reviews, "model development" brings ambiguity to what is described. In the majority of cases "model development" means you worked and modified the source code of HEC-Ras in a certain way, if this was not the case I would recommend to change it to "set-up" or "build".

p.6 line 7. "Computationally efficient" needs references on other studies which prove the point.

p.12 line 18. Information missing on satellite image source and the way it was processed.

p.12 line 25. "However, *when* the flood level …"

p.12 line 30. "Further research …" such statements should be moved to the Conclusion section.

p.13 line 8. The reference (Helm, 1996) doesn't belong to the statement. The study of Helm gives the definition of the risk used in this manuscript at line 11. Klijn (2009) and Van Manen and Brinkhuis (2005) used already existing definitions for their research. This has to be corrected.

p.14 line 27. The indicator of error has to be specified.

p.20 line 17. "poor" is a rather vague definition, some numbers or indicators would bring more clarity.

p. 21 line 2. "risky" is recommended to be removed.

p.22 line 20. "Probabilistic flood maps…" is not clear, recommended to be rephrased.

p.23 line 5 "…, the data provided…" this part of sentence should be rephrased.

---

## Referee Report (RR3)

**Review of manuscript:** "*Flood risk assessment due to cyclone induced dike breaching on coastal areas of Bangladesh*".

**Overview:**

The paper has further improved from the previous version, and provides a clear explanation of its methodology and case results. The abstract, introduction and conclusion also now ensure the work is given better context for international readers. The authors should also be commended for their thorough responses to the issues raised, which makes subsequent reviews much less demanding.

The general comments I had previously made (methodology, message and English) have all been addressed, as well as all the specific comments. I therefore suggest the paper to be 'accepted subject to minor revisions', suggestions for which are given below. While most are suggestions or very minor mistakes, the issue with the HEC-RAS boundary conditions is important and should be considered.

**Comments:**

**Abstract**

- P1, Line 25: CycloneS
- P1, Line 27: LocationS

**Introduction**

- P3, Line 15-20: This seems to be a repeat of the abstract. Consider reducing or rewriting.

**Study Area**

- P2, Line 1: 80% of the polder is at 1.55m? I assume the authors mean 80% is above/below 1.55m or that 80% of the polder has an elevation of about 1.55m. Please clarify.
- P2, Line 2: MSL: the term has not been used before, please add (mean sea level)
- P2, Line 6: Do these figures relate to this polder specifically? I.e. did 94 people die in this polder?

**Methodology**

- P7, line 3: "…using discharge as the west boundary and water level as the east boundary conditions". Why is this done? It seems that the sea level (i.e. water level boundary condition) will dominate the entire stretch on both 1D sections, and using Q as a BC just complicates things.
- P7, Line 5: "HEC- 5 RAS generates mesh with irregular shapes." I think this can be removed, or changed to 'meshes'.
- P13, line 4: "The adapted depth-damage curves are obviously simplistic ones.". I don't think this is needed

**Conclusions**

- P22 Lines 22 – 26: For me there is no need to use the values here, they are already explained above.
- P22 Line 23: Comma and full stop after equal

---

## Author Response (AR4)

**Round #1**

**Response to review comments**

Manuscript title: Flood risk assessment due to cyclone induced dike breaching on coastal areas of Bangladesh
Manuscript Id: NHESS-2018-169

**Review comments of Reviewer # 1**

*Comment*:

Methodology: The methods used for calibration, breaching analysis and scenario development are all unclear and open to debate, but the biggest problem is the 1D2D model used. It is not described clearly, and as I understand it, models the sea in the 1D component. If this is the case, it requires a much better explanation and/or figures.

> *Authors' response:*
>
> *Thank you for the comment. The revised manuscript provides more detailed information in the methodology section, by adding a flow chart to bring more clarity, following your suggestion. The methodology section has been improved in adding explanations on data, model development, scenario development and maps.*

Message: I find the message of the paper ambiguous. The discussion and start of the conclusion mention the dynamics of the case-study, which make sense as discussion topics. However, conclusions about flood forecasting and early warning systems seem out of place. Perhaps the potential development of PFMs for other polders is better suited to be discussed in the conclusions. Lines such as 'end the problem of poverty' should certainly be reconsidered (discussion and conclusion section has been adjusted to provide more focused and clearer message, limitations of the research and scope of future research).

> *Authors' response:*
>
> *Please see the revised version of the Conclusion in the new submitted manuscript. Authors hope that it is more coherent and connected to the sections of the paper..*

English: Multiple mistakes are found which distract the reader and give the impression of a careless approach to the work. In the specific comments below, only the ones found in the introduction are listed, but many more exist throughout the manuscript (discussion and conclusion have been adjusted according to the research objective and provide a coherent message).

> *Authors' response:*
>
> *To improve the level of English, native English speakers have been consulted and the revised version of the manuscript has been adjusted following their suggestion.*

**Reviewer 1:**
**"**P1, Line 22: Presumably this second smaller abstract is not meant to be part of the main abstract. This is perhaps a formatting error during the upload process.**"**

*Authors' Response:*

*Thank you for pointing this out. The error been corrected and the manuscript has been adjusted (please see the revised abstract).*

**Reviewer 1:**
**"**P2, Line 7: "…to protect the land from flooding due to diurnal high tide". The English here is incorrect. Either 'high tides' or 'the diurnal high tide'.**"**

*Authors' Response:*

*To improve the level of English, native English speakers have been consulted and the revised version of the manuscript has been adjusted following their suggestion (on page 2, line 4).*

**Reviewer 1:**
"Also, do the polders not also protect from the heavy rainfall mentioned just before this sentence?**"**

*Authors' Response:*

*Most of the polders of Bangladesh were built under the project titled as Coastal Embankment Project (CEP) in the 1960s. However, several articles and reports such as Mondal et al., 2006, Islam 2006, Islam et al., 2013, Bangladesh Delta Plan 2100, Coastal Embankment Improvement Project (Phase I) (Main Report) etc. indicated that the polders were constructed to protect the land from tidal flooding and salinity intrusion.*

**Reviewer 1:**
"P2, Line 14: "Rising the crest level...". English.**"**

*Authors' Response:*

*As mentioned previously English has been checked for improvement.*

**Reviewer 1:**
"P2, Line 16: "Effect of…". English**"**

*Authors' Response:*

*English has been checked and corrected .*

**Reviewer 1:**
"P2, Line 16: "Moreover, non-structural flood mitigation measures such as (…) and (…) is currently unavailable for the coastal areas of Bangladesh". English.**"**

*Authors' Response:*

*English has been checked and corrected following the suggestion of native English speakers.*

**Reviewer 1:**
"P2, Line 23: "Furthermore..". This sentence suggests that SLR is not an effect of climate change. Did Mendelsohn et al. include this in their study?**"**

*Authors' Response:*

*Authors acknowledge that the Sea Level Rise is indeed result of climate change and the paragraph was rearranged and adjusted to clarify the message. Moreover, English was checked and corrected as needed (on Page 2, line 21 to 25).*

**Reviewer 1:**
"P2, Line 31: I cannot find where the variables of breach width, height and propagation are analysed in the study**"**

*Authors' Response:*

*As the reviewer indicated in the later comments that the breach properties were not independent variables, it should not be stated as a parameter of scenario development. Authors acknowledge the suggested correction and the revised version was adjusted accordingly (on page 2, line 30 to 33).*

**Reviewer 1:**
"P2, Line 31: Scenarios mentioned only previously in abstract. Authors could consider a minor revision here**"**

*Authors' Response:*

*Authors acknowledge the suggestion and manuscript was adjusted accordingly (on page 2, line 27 to 33).*

**Reviewer 1:**
"P3, Fig. 1: Upazilla term used in figure, but not explained in text. Presumably it is a form of district**"**

*Authors' Response:*

*Thank you for raising this issue the manuscript has been adjusted accordingly (on page 3, line 5).*

**Reviewer 1:**

"P3, Fig. 1: Would it be possible to indicate the extent of the mangrove forests?**"**

*Authors' Response:*

*The maps were adjusted and the extent of the mangrove forest was added in the maps (figure 1, 3, 8, 12 and 13).*

**Reviewer 1:**

"P3, Line 7: Who has classified this? The authors or a governmental body?**"**

*Authors' Response:*

*The classification was done by the Ministry of Land of Bangladesh. The information was added in the revised version (page 3, line 8).*

**Reviewer 1:**

"P4, Line 18: "…simulated using discharge as the upstream boundary…". What discharge? Is it important? Is it correlated to the cyclonic rainfall? Is it negligible in relation to the water level.**"**

*Authors' Response:*

*The authors were trying to describe the developed 1D model which was calibrated against the measured water level and discharge on the river stations and then couples with 2D model. The 1D model was calibrated for normal condition (without cyclones) as hydrometric data during cyclonic events were not available. Moreover, the simulated cyclonic event "Sidr" made landfall in the coast of Bangladesh during the month of November which is post monsoon season. In the past, most of the cyclones his the coast of Bangladesh during the months of April-May or October-November which are pre and post monsoon where discharge from rivers don't play a significant role. The authors acknowledge this and the "Methodology" section was adjusted to provide more detailed and clearer description.*

*To make it more clear, flow chart of activities performed (Figure 2), 1D model network, 2D mesh extent and resolution in the map (Figure 3), sample cross sections (Figure 4 and 5), water level on the sea side network (Figure 6) has been added. More about modelling tool (page 6, line 1 to 9), data used (Table 1) and explanations about the model were added.*

**Reviewer 1:**
"P4, Line 29-30: "…and the location furthest from the dike breach is most sensitive." Given we don't (yet) know the locations of the breach or in which direction from the breach you mean, this is very ambiguous. You presumably mean in areas of low flow.**"**

*Authors' Response:*

*As suggested by the reviewer, authors' intention was to indicate about the sensitivity of the low flow areas to the coefficient of roughness "n", indeed and the manuscript was adjusted accordingly by moving the corrected section to page 14, line 29.*

**Reviewer 1:**
"P5, Line 3: This paragraph about data gathering seems out of place, considering that data gathering was described before the previous paragraph about sensitivity analysis.**"**

*Authors' Response:*

*Authors adjusted accordingly in the reviewed version and moved to page 5, line 14 to 19.*

**Reviewer 1:**
"P5, Line 6: Perhaps you should mention that the flood extent data from MODIS data was (presumably) used for calibration.**"**

*Authors' Response:*

*The manuscript was adjusted accordingly (on page 5, line 18, 19 and on page 14, line 23, 24).*

**Reviewer 1:**
"P5, Fig. 2. Please indicate the Khaprabhanga river on the map**"**

*Authors' Response:*

*The maps were adjusted accordingly (figure 1, 3, 8, 12 and 13).*

**Reviewer 1:**
"P5, Fig. 2. As I understand it, the 1D component of the model stretches right around the polder, from the start of the Khaprabhangra river into the foreshore. Can you indicate the extents on the map?**"**

*Authors' Response:*

*The 1D components were added in the map (figure 3).*

**Reviewer 1:**
"P5, Line 12: "For the rivers, the surveyed cross…". You are presumably referring to the Andharmanik and Galachipa rivers on the east and west sides of the polders, but the previous sentence mentions only river. Please clarify this.**"**

*Authors' Response:*

*The revised version of the manuscript was adjusted to have more clarity (on page 7, line 12 to 16).*

**Reviewer 1:**
"P5, Line 13: I find the use of 1D channels to simulate the foreshore very irregular, and feel it deserves more explanation or references of previous methods. Are these channels connected to the river channels? Is discharge a factor? It is not mentioned."

*Authors' Response:*

*Thank you for the comment. The authors tried to represent the condition of the water bodies adjacent to the dikes with 1D model with synthetic boundary condition. As the coast of Bangladesh is flat and shallow, a 2D model for coastal hydrodynamics will require inclusion of larger area of the sea which was not the area of interest and which would have increased the simulation time too. The utilised 1D network was not connected with the rivers as explained earlier the river water didn't play a major role during the previous cyclones as it's pre or post monsoon. The authors acknowledge this requires clearer description. The revised version was adjusted to provide more detailed and clearer explanation (on page 7, line 12 to page 8, line 2).*

**Reviewer 1:**
"P5, Line 14: 13 control structures are mentioned here, which are presumably the 'Sluice gates' indicated on the map. If they are, please use the same term, and also, why are 13 not indicated?**"**

*Authors' Response:*

*The map was adjusted accordingly (figure 3).*

**Reviewer 1:**
"P5, Line 16: "Therefore, the canal network inside the polder was not included in the 1D model". The canals will have no effect on the dynamics outside the polder, but once flooding occurs they almost certainly will. I understand the DEM resolution will be too coarse to capture them, but this fact should be mentioned.**"**

*Authors' Response:*

*The canal network inside the polder was not connected to the river network as during a cyclone the gates of the control structure will remain closed. But the larger canals were included in the DEM. The width of the larger canals were wide enough to be included in the DEM. This was explained in page 8, line 3 to 7.*

**Reviewer 1:**

"P5, Line 19: Surely the foreshore data has no average slope?**"**

*Authors' Response:*

*The authors agree that the rivers had higher slope than the foreshore area. As the distance between computational points is inversely proportional to slope, the rivers will require smaller Δx and same Δx will reduce instability for the foreshore area too. A clearer explanation was provided in the revised version (on page 10, line 2 to 4).*

**Reviewer 1:**

"P6, Line 13: Are these storm surge heights directly applied as boundary conditions to the 1D model at every 1D cross-section location on the foreshore. I find this very difficult to understand.**"**

*Authors' Response:*

*To ensure same water level at all the points of the foreshore reach, same water level was applied at both ends of the reach as hydro dynamic boundaries. The authors acknowledge this requires clearer description. The revised version was adjusted (on Page 7, line 17 to page 8, line 2 and figure 6).*

**Reviewer 1:**

"P6, Line 16: This seems out of place, perhaps more suited to the literature review earlier."

*Authors' Response:*

*Thank you for the suggestion. Agreed that part of the paragraph can also be suited in the literature review section and part of the paragraph  was moved to page 10, line 23 to 28.*

**Reviewer 1:**

"P6, Line 26: Where is this section? As mentioned it should be in the map"

*Authors' Response:*

*Following your suggestion, the extent of mangrove forest was depicted in the maps (figure 1, 3, 8, 12 and 13).*

**Reviewer 1:**

"P6, Line 30: It was previously indicated that the breach geometry and propagation were variable in the scenario make-up (Abstract and Introduction). However in the end they are dependent on the other variables. This should be made clear"

*Authors' Response:*

*Indeed these are not independent variables and should not be stated as parameters for scenario development and the revised version was adjusted accordingly (on page 10, line 32).*

**Reviewer 1:**

"P7, Line 6: Why not call the scenarios east west and central for simplicity?"

*Authors' Response:*

*Thanks for the suggestion. However, the change of the tittle of the scenarios will require the adjustment of most of the figures and tables. Therefore, no change was made.*

**Reviewer 1:**

"P7, Table 1: The SLR variation is based on current conditions and a possible future rise in 2100. This raises the question as to which period the PFMs that have been developed correspond. Perhaps it makes more sense to vary SLR for a given future moment according to the RCP scenarios. Also, the 1/25yr surge height used for the cyclone is presumably for current conditions, but as you explain earlier, this is subject to change."

*Authors' Response:*

*Thank you for your suggestion. As stated earlier SLR has been considered according to the RCP scenarios. The storm surge height for 1/25 yr event was based on current condition. As not enough measured data was available, previous literatures were used for determining the storm surge height for simulation. Time series data for sea level rise for the study area was not available either. Therefore, the scenarios were developed for the years for which data was available. The section was described more clearly in the revised version (on page 10, line 16-17).*

**Reviewer 1:**

"P7, Line 9: I don't understand this. If flooding results from the 3 worst case scenarios are available, it surely means breach locations are already selected. So how does this allow for a critical breach location to be selected? Is this flooding from overtopping of the dikes?"

*Authors' Response:*

*The flooding occurred for breaching only as the crest level of the dike was considered to be elevated to a height suggested by CEIP in the future. Breaching of the dike was considered for all the scenarios, therefore, authors agree that dike was breached already in the three worst case scenarios as well. The intention was to present a methodology to compare and identify a critical location based on damaged caused by flood in case of breaching of dike which could be applied in other locations. Providing better protection at the critical locations might reduce the damage significantly during and after a cyclone. The authors adjusted the revised version to provide clearer explanation (on page 11, line 3 to 5).*

**Reviewer 1:**

"P8, Line 3: "…depth-damage curves from elsewhere." This is explained later, but at this point the sentence is very ambiguous."

*Authors' Response:*

*It was adjusted accordingly (on page 12, line 9 and 10).*

**Reviewer 1:**
"P9, Line 5: "The critical location of breaching…" Why is this included here? It adds to my existing confusion about how these locations are selected."

*Authors' Response:*

*It was adjusted accordingly (page 13, line 3 to 5).*

**Reviewer 1:**
"P10, Line 2: "we have assumed that the probability of occurrence of the hazard and the probability of failure of dike as the same". Can the authors estimate the accuracy of this assumption? Presumably no flooding occurs (from overflow) of the dike in the simulations without breaching, but perhaps wave overflow would occur?"

*Authors' Response:*

*The crest level of the dikes was considered to be at the design height suggested by Islam et al., 2013 and CEIP. Wave action was considered during the calculation of design crest level and a free board was also considered. It can be safely concluded that the crest level suggested by Islam et al., 2013 will be sufficient enough to protect area inside the dike from overtopping during a 1/25 year event. Moreover, the breaching of a dike depends on the physical condition of the dikes and it's soil properties. Neither of these data were available. Therefore, the calculation of probability of the dike breach was not possible. To simplify and to investigate the effect of dike breach, the breaching probability was considered same as the cyclonic event. This was described more in the revised version (page 11, line 2 to 5 and page 13 line 15 and 19).*

**Reviewer 1:**
"P10, Line 4: I don't understand the relevance of this reference, as all scenarios used in the study have the (assumed) same probability"

*Authors' Response:*

*The authors intention was to provide reference for the equation used for calculation of probabilistic flood maps. It was rephrased in the new version of the manuscript, for better clarity (page 14, line 3 and 4).*

**Reviewer 1:**
"P10, Line15: "…comparing the observed and simulated water level and discharge". As mentioned previously, you have not mentioned what discharges are being simulated, or what you are calibrating them to. Also if the cyclone water levels are applied as boundary conditions, surely the calibration is trivial?"

*Authors' Response*

*As mentioned earlier the calibration of 1D model was done for normal condition, not for a cyclone event as no data was available during that event and very few data was available for calibrating the 2D model. The intention of the authors was to present a methodology with which flood risk maps and*

*PFMs can be generated for different locations. More detailed description was provided in the revised version (on page 6, line 16 and 17and page 7, line 21 to page 8, line2).*

**Reviewer 1:**
"P14, Figure 7: Can the authors explain why the damage decreases for larger flood depths?"

*Authors' Response*

*The damage is a function of flood depth but the unit is per unit area(per m$^2$). Therefore, if the flood extent for higher depth is lower, the damage due to flood might be lower too. It was explained in the revised version (on page 21, line 6 and 7).*

**Reviewer 1:**
"P15, Line 6: Ignoring depths less than 0.5m seems quite extreme, can the authors explain why this was done?"

*Authors' Response*

*The developed damage curves suggest that the damage for flood depth below 0.5 m is minimal. Moreover, the authors tried to explore the effect of living with flood concept. Also tried to consider the uncertainty of the DEM and the 2D inundation model. The depth as 0.5m is an arbitrary depth. For a country where flood is a recurrent phenomenon with larger depth, living with flood might be already adopted by the local people. This was explained in the revised version (on page 19, line 6 to 11).*

**Reviewer 1:**
"P16, Line 22: "Figure 4 demonstrates that the depth of flooding gradually decreases as the water moves inland". This is not true, the figure only shows inundation extent. Perhaps the authors mean imply."

*Authors' Response*

*Authors agrees and thanks the reviewer for identifying and was adjusted accordingly by removing the statement.*

**Reviewer 1:**
"P17, Line 18: "…(Fig. 8). Therefore, although canals play a crucial role in the economy and social life of the area, they also increase the risk of flooding". This is a strange, and in my view, inaccurate conclusion. Figure 8 shows the residential areas as high risk because they are more valuable. They happen to be situated beside canals."

*Authors' Response*

*Indeed the residential areas had higher depth damage ratio than other land classes. But the residential areas were flooded primarily for being by the side of the canal. The authors' intention was to state that as these areas are adjacent to the canals for various reasons such as being advantageous for transportation, also makes them susceptible to flooding and higher damage. The authors will try to explain more clearly in the revised version (page 21, line 23 and 24).*

**Reference**

Mondal, M.K., Tuong, T.P., Ritu, S.P., Choudhury, M.H.K., Chasi, A.M., Majumder, P.K., Islam, M.M. and Adhikary, S.K., 2006. Coastal water resource use for higher productivity: participatory research for increasing cropping intensity in Bangladesh. *Environment and Livelihoods in Tropical Coastal Zones: Managing Agriculture-Fishery-Aquaculture Conflicts.*, pp.72-84.

Islam, M.R., 2006. 18 Managing Diverse Land Uses in Coastal Bangladesh: Institutional Approaches. *Environment and livelihoods in tropical coastal zones*, p.237.

Islam, M.S., Alam, R., Khan, M.Z.H., Khan, M.N.A.A. and Jahan, S.N., 2013. Methodology of crest level design of coastal polders in Bangladesh. In *4th International Conference on Water & Flood Management*.

**Review comments of Reviewer # 2**

Review of manuscript: *"Flood risk assessment due to cyclone induced dike breaching on coastal areas of Bangladesh".*

**Overview**

The paper describes the methods and suggests tools for the probabilistic flood mapping in a polder area of Bangladesh. The study area selected by the author is interesting in terms of its geographical complexity and challenges related to the data collection. The methods used are rather simplified and aimed at giving a general overlook on the problem.

*Authors: Thanks for the comments*

**The main concerns**

There are however, some major concerns about the idea behind methods and scenarios selection. The research questions should be addressed in Discussion section. One of the main problems is the description and structure of Study area and Methods sections. Some additional references are required in places where it is not clear where exactly the data or information come from. In addition, there is a large amount mistakes in language usage, both grammar, punctuation and word selection. The figures are not consistent throughout the manuscript. Therefore, my recommendation is to return this manuscript to authors for major revisions.

*Authors: We acknowledge the concerns raised by the reviewer. The manuscript was revised and adjusted to bring more clarity in the description of the study area and the presented methodology (see section "Study area" and "Methodology". The discussion section was revised as well. More specific answers were provided in the following sections (the manuscript was revised to present more clear objectives, coherent findings and innovation of the research).*

**General comments**

**Reviewer 2:**

Study area. Due to the specific conditions of the region, it is important to give more clarity and structure to this section. Probably it is a good idea to consider removing some unnecessary information and add more visualisation to more important aspects that are crucial for this specific research.

*Authors: The authors acknowledge the suggestion. More information about the study area was provided to improve the description of the study area.*

**Reviewer 2:**

Research question. Should be stated clearly what exactly is developed within the study and to which degree it is considered innovative.

*Authors: The authors understand that the innovation remained unclear in the draft. The main innovation of the research was presented clearly in the revised manuscript (The introduction and Conclusion section have been adjusted to present the research questions and innovations).*

**Reviewer 2:**

Methodology. Here are major rewritings are required to increase the quality of the paper. Modelling sub-section needs more clarifying in tools selection and usage. In addition, I suggest more description of the data used for model set-ups and calibration.

*Authors: The authors acknowledge the suggestion. More detailed information was provided in the methodology with a flow chart to bring more clarity in the revised version. The description of the data used for the setting up of the model and its calibration were elaborated (Methodology has been revised by adding a flow chart of activities performed (figure 2), more about the software and data used (page 6, line 1 to 9, table 1), example of cross sections used (figure 4 and 5), the extent of 1D network and 2D mesh (figure 3), simulated water level on the sea side and adjusted write up).*

**Reviewer 2:**

The subsection 3.2 Cyclonic scenarios considered; the selection of the values for different scenarios based on the IPCC report is rather subjective. It is suggested to consider regional sea level changes rather than global mean, as there is a significant difference specifically for Bangladesh. This may bring more impact on the outcomes of the study.

*Authors: The authors' intention was to present a global methodology applicable to everywhere. The relative sea level rise for RCP 8.5 of IPCC AR5 for the coast of Bangladesh is 0.56 m (GERIC, 2015) which is slightly lower than what was considered by the authors (0.63 m).*

**Reviewer 2:**

Discussion and Conclusion. It would be worth writing how/if the future studies would improve the current outcomes.

*Authors: The authors acknowledge the suggestion. The manuscript was revised and recommendation for future studies was added as well. (please see "conclusion" section).*

**Reviewer 2:**

The take-home message is rather vague. The discussion section needs major re-writing in accordance to the research questions stated in Introduction. In my opinion such general methods used in this study should be accompanied with rather more detailed (sub)-section on the sources of errors and limitations.

*Authors: The discussion section was adjusted with the reflection on the research questions. Limitations of the study were added in the conclusion (please see "Discussion" and "Conclusion" section).*

**Reviewer 2:**

Heroic assumptions such as "lead to economic growth" and "end the problem of poverty" should be avoided.

*Authors: The authors acknowledge the suggestion and such phrases were removed.*

**Reviewer 2:**

English. A serious revision of the language is necessary to improve the quality and readability of the manuscript. Among main issues I would outline: plural vs. singular, passive voice use, punctuation,

repetitions of the same structures in consecutive sentences/paragraphs, repetitions of abbreviation explanations, articles selection, language use, etc.. The specific remarks do not cover language issues.

*Authors: To improve the level of English, native English speakers have been consulted and the revised version of the manuscript has been adjusted following their suggestion.*

**Specific remarks**

**Reviewer 2:**

p.2 line 2. According to Neumann et al (2015) 49% of population located in low elevated coastal zone for the year 2000, at that time the overall population of Bangladesh was 139 mil. Values should be corrected.

*Authors: The authors acknowledge the suggestion and the manuscript was adjusted accordingly (on page 1, line 26 and 27).*

**Reviewer2 :**

p.2 line 11. The number US$1.67 million seems rather small, needs additional check.

*Authors: The authors thank the reviewer for pointing it out. The revised manuscript was corrected accordingly (on page 2, line 7).*

**Reviewer 2:**

p.2 line 14. "Raising the crest level …" the sentence is unclear.

*Authors: Agreed and revised (on page 2, line 12).*

**Reviewer 2:**

p.2 line 15. References needed to indicate which exactly previous studies were done in this matter.

*Authors: The authors acknowledge the suggestion and the revised manuscript was adjusted accordingly (on page 2, line 13).*

**Reviewer 2:**

p.2 line 17. It needs more clarification how land use zoning address the flood mitigation.

*Authors: The authors think that land use zoning is widely used as a flood risk mitigation measure. Reference was added to bring more clarity (on page 2, line 18).*

**Reviewer 2:**

p.2 line 19. "…of these tropical cyclones will increase…" the statement will is rather confident, however it is likely increase. We are not 100% sure it will increase the intensity of storms. Look further through the manuscript for same errors.

*Authors: Indeed, it was adjusted accordingly (on page 2, line 19)."*

**Reviewer 2:**

p.2 line 26. Which exactly severe consequences specifically in Bangladesh? Look at Neumann et al (2015) for ideas.

*Authors: As suggested, adjusted in the revised version.*

**Reviewer 2:**

p.3 line 3. It is recommended to visualise coordinates in Figure 1.

*Authors: Coordinates were added in Figure 1 (figure 1).*

**Reviewer 2:**

p.3 line 6. The source of census data is missing.

*Authors: We thank the reviewer for pointing it out. Source was added in the revised version (on page 3, line 6).*

**Reviewer 2:**

p.3 line 12. Consider the importance of putting the local names of seasons to the manuscript.

*Authors: The local names of the copping seasons have been mentioned. However, the calendar months are mentioned as well and therefore, Authors think that any lack of clarity is not obvious.*

**Reviewer 2:**

p.3 line 15. Some figures on the land subsidence rates may bring more light on the severity of the problem in the region.

*Authors: Value for land subsidence for the region was searched and added (on page 3, line 16).*

**Reviewer 2:**

p.4 line 12. "Model set up" rather than "model development"

*Authors: Thank you for the suggestions. As we were unable to identify the mistake, native English speakers were consulted for clarity and following there suggestion, no further change was made.*

**Reviewer 2:**

p.4 line 20. The reference on FINMAP is missing.

*Authors: Reference for FINMAP was added (page 5, line 10).*

**Reviewer 2:**

p.4 line 26. More details on the computation mesh are recommended.

*Authors: This was explored and more detailed was added (on page 7, line 4 and 5).*

**Reviewer 2:**

p. 5 line 9. The version of the model is missing.

*Authors: We have used version 5 of the HEC-RAS tool. This information was added in the revised manuscript (on page 6, line 1).*

**Reviewer 2:**

p.6 line 17. I would include the figures on the land subsidence.

*Authors: Value for land subsidence for the region was searched and added (on page 3, line 16).*

**Reviewer 2:**

p.6 line 23. The figures of SLR indicated could be updated to the ones for 2100.

*Authors: The Authors' intention was to state the probable sea level rise suggested by IPCC. The authors took a look and no further adjustment was not required.*

**Reviewer 2:**

p. 7 line 15. It is better not to describe indirect damages if they are not consider further.

*Authors: We have not elaborated on indirect damages. We have just defined it to bring clarity.*

**Reviewer 2:**

p.8 line 16. More reasoning for choosing of the figure of 50% would bring more light on the selection.

*Authors: Muktadir and Hasan, 1985 stated in their study that the rural house hold of Bangladesh usually are built around a large country yard. The land classification by the Ministry of Land defines the whole house hold including the country yard as residential area. But the damage curve considered for the residential area was for the damage to the house and the country yard will not have significant damage if flooded. Considering this the authors tried to exclude the country yard from the residential area. The satellite image of the area indicated that the about half of the area of residential complex (house hold) is usually empty. Therefore, the figure 50% was considered. This was added in the revised version (on page 12, line 15 to 19)*

**Reviewer 2:**

p.9 line 7. "More research…" is rather suitable for conclusion.

*Authors: Authors acknowledge the suggestion and the manuscript was adjusted accordingly by removing the statement.*

**Reviewer 2:**

p.9 line 16. This definition of risk was presented earlier by Helm 1996. See Helm, P. (1996). Integrated Risk Management for Natural and Technological Disasters. Tephra, 15(1), 4-13.

*Authors: Authors acknowledge the suggestion and the manuscript was adjusted accordingly (on page 13, line 8).*

**Reviewer 2:**

p.10 line 9. It is not clear where M and N are in your formula.

*Authors: The parameters M and N were defined in the manuscript following the equation. The equations were adjusted to bring more clarity (on page 14, Equation 5).*

**Reviewer 2:**

p.11 Figure 4. The boundaries are not clear, some simplification of shapes could bring more readability to the map.

*Authors: Authors acknowledge the suggestion and the map was adjusted to increase readability (figure 8).*

**Reviewer 2:**

p.12 Figure 5. There is some confusion what exactly this figure is supposed to show.

*Authors:*

*Authors' intention was to depict the effect of different variables of the scenario development. For example: to depict the effect of sea level rise, two scenarios with keeping all the other variables (except from sea level rise) constant and changing the SLR, were compared. Authors acknowledge that this requires clearer explanation and the revised manuscript was adjusted (on page 20, line 3 and 4).*

**Reviewer 2:**

p.15 Figure 8. The map layout is not consistent with other maps.

*Authors: The maps were adjusted to have consistent layout (figure 3, 8, 12 and 13).*

**Reviewer 2:**

p.15 line 6. Some elaborate clarification why 0.5 m is used. My guess, some damages might be underestimated by selecting such high value.

*Authors: The developed damage curves suggest that the damage for flood depth below 0.5 m is minimal. Moreover, the authors tried to explore the effect of living with flood concept. Also tried to consider the uncertainty of the DEM and the 2D inundation model. The depth as 0.5m is an arbitrary depth. For a country where flood is a recurrent phenomenon with larger depth, living with flood might be already adopted by the local people. Moreover, this arbitrary was used for PMFs and the estimation of damage due to flood was not affected. This was added in the revised manuscript (on page 19, line 7 to 11).*

**Reference**

Climate Service Center Germany (GERIC) and KfW Development Bank, 2015, CLimate Focus Paper: Regional Sea Level Rise South Asia, Hamburg and Frankfurt am Main.

Muktadir, M.A. and Hasan, D.M., 1985, December. Traditional house form in rural Bangladesh: a case study for regionalism in architecture. In Regional seminar on Architecture and the Role of Architects in Southern Asia (pp. 19-23).

**Round #2**
**Response to review comments**

**RESPONSE TO EDITOR'S COMMENTS**

***Comment***: Both reviewers have looked at the revised manuscript and find that it has considerably improved. However, they also provide a number of comments where they think that improvements are still necessary. I have looked as well at the manuscript and think that clarity and conciseness should further be improved. Moreover, I miss clarification about my earlier remark on the first version: "... It also needs to be very clear what the scientific contribution and innovation of the work is. Case studies can be published in nhess, however, it does not suffice to only apply established methods, so you should be very careful in pointing out what could be transferred, whether there are generic findings etc...." Please think about what this work could offer to the international audience of nhess that is not interested in your case study area, but could be interested in whether/how your insights and methods could be transferred to other area. It would be good to have a few thoughts/sentences in the introduction, discussion and/or conclusions chapters on the issues of scientific contribution and transferability.

***Reply***:

*The authors appreciate your comments. We have updated the manuscript to reflect the applicability at other location so that the manuscript does not read like as a case study report.*

Specific changes made to the manuscript are:

*In Abstract (P 1, Line 25 to 30):*

"The frequency and intensity of the cyclone around the world are likely to increase due to climate change which will require resource intensive improvement of existing or new protection works for the deltas. Identification and prioritising maintenance of critical location of dike breaching can potentially prevent a disaster. Non-structural tools such as land use zoning with the help of flood risk maps and probabilistic flood maps has the potential to reduce risk and damage. The method presented in this research can potentially be utilized for the deltas around the world to reduce vulnerability and flood risk due to dike breach cause by cyclone induced storm surge."

*In Introduction (P3, Line 15 to 20):*

"With the effect of climate change, the frequency and intensity of the cyclone around the world will increase. The protective structures for the deltas around the world will require adjustments which will be resource intensive. Non-structural tools such as land use zoning with the help of flood risk maps and probabilistic flood maps has the potential to reduce risk and damage. Identification of critical location of breaching and intensifying the maintained effort for these locations can potentially prevent a disaster. Although coastal region of Bangladesh was selected as the case study area, the method presented in this research can be utilized for the vulnerable deltas around the world."

*In Conclusion (P 24, Line 1 to 6):*

"Climate change will likely cause increase in the frequency and intensity of the cyclone around the world. This will call for large investments for the improvement of existing or new protection works for the deltas. Identification and prioritising maintenance of critical locations of dike breaching can potentially prevent a disaster. Non-structural tools such as land use zoning with the help of flood risk maps and probabilistic flood maps have the potential to reduce the risk and the damage due to dike breaching. The method presented in this research can potentially be utilized for the deltas around the world to reduce vulnerability and flood risk due to dike breach caused by cyclone induced storm surges."

The authors are very thankful to the Editor for the valuable comments. Addressing the comments has helped us in improving the manuscript.

**RESPONSE TO COMMENTS OF REVIEWER # 1**

*Comment*: **Overview:**

The paper has improved significantly in all of the three main points mentioned previously (message, methodology and English). This, along with the improved figures, gives the reader a much better understanding of the nature of the research. However, the problems with these areas have not been fully resolved. Despite the much improved manuscript, in my opinion these issues still require major revisions to the methodology and the message. However, methodological changes or extra data analyses are not necessarily needed to achieve a publishable article.

*Reply: We are thankful to the reviewer for the appreciation and in the following we provide answers to the specific review comments.*

*Comment*: **General Comments:**

The improvements from the previous manuscript, as well as the continued (or new) problems I observe in the 3 areas mentioned above are described below;

**Methodology:**

The methodology is much clearer in the second manuscript, and helped by the flowchart. However, there are 3 aspects that need to be addressed.

*Reply: Specific replies are provided below against specific comments*

*Comment* [Methodology, part 1]: The first is the differentiation between the 3 extreme scenarios the 72 scenarios. I previously suggested that a different name be given for these sets of scenarios, but this was not done, and it requires careful analysis for the reader to dissect this for themselves. The flowchart helps a little but it also causes more questions, which are listed in the specific comments below.

*Reply: The authors apologise for not incorporating this suggestion in the previously revised manuscript. Figure 2 (shown below as well) has been updated now to incorporate this suggestion.*

[Figure]

Figure 1: Methodological approach followed in this study.

*It may be noted that the scenarios have been named as S1, S2 and S3. Subsequent discussion in the manuscript (for example, P 11, L9) have been modified to include the names of these three scenarios.*

***Comment*** [Methodology, part 2]: The second aspect is the fact that all the outputs relate to a 1:25yr event. Highlighting this more clearly would help the reader to understand the context of the research in terms of a wider flood risk analysis for the region, (see comments on message, below). A serious problem not addressed with the results of this calculation event is that the authors do not define for which temporal period risks are applicable. For example, if the results represent the current risk, then of the 72 simulations, those simulations with future SLR should not be included. For the risk in 2100 (the date at which the SLR value is predicted to happen), the simulations with current sea level should not be used. Furthermore, subsidence over time was accounted for to estimate the current topography, but not future topography. This is briefly mentioned in the conclusions, but not the significance for the results are not.

***Reply***:

[Part a, 1:25-year event] *Authors appreciate the suggestions and we have updated the manuscript to reflect this aspect. As shown below we have included this explanation in the introduction, discussion and conclusions sections.*

*P 1, Line 15:* "Storm surge for a cyclone event of 1 in 25 return period was considered for all the scenarios."

*P 2, Line 17 and 18:* "A storm surge event of 25 year return period was considered in this study for the generation of different scenarios."

*P 3, Line 14 and 15:* "A cyclone event of 25 year return period was considered in this research."

*Furthermore, the following explanation on the results of scenarios with and without the climate change (sea level rise) impact have been added.*

*P 11, Line 12 to 13:* "As the storm surge height suggested by Islam et al. (2013) corresponds to an event of 25 year return period, the PFM generated in this study corresponds to 1 in 25 year return period."

[Part b, temporal period]

*This is indeed an important point and we have added the following explanation in the discussion chapter:*

*P 2, Line 22 to 27:* "The scenarios with the effect of climate change (sea level rise) had more damage compared to the scenarios without climate change. Scenarios S1, S2 and S3 were associated with the highest storm surge height. With the same set of conditions without climate change (sea level rise) the storm surge height was 6.52 m PWD (Table 2). The damage corresponding to breaching of eastern, central and western locations of the dike due to the storm surge without climate change impact were 23.3%, 20.5% and 21.7% respectively lower than the damages with the climate change. The corresponding values for the flood extent were 30.1%, 21.67% and 27.21% respectively lower than the flood extent areas with the climate change impact."

*P 22, Line 29 and 30:*

"The scenarios considering the effect of climate change (sea level rise) indicated that the flood extent and damage due to flood will increase with sea level rise."

[Part c, subsidence]

This indeed is an important point and it has been explained with the following text (P5, Line 13 to L 17):

> "The same DEM was used for the simulations of the year 2100 without any corrections for further subsidence. Subsidence of the coast in the past has been reported by Brown and Nicholls (2015) and was verified with the survey data from IWM and FINMAP. Subsidence in the future may continue but in the absence of scientific studies it was not considered for the future scenarios in this research. It is

noteworthy that if subsidence continues then the effect of the SLR may be increased and the results reported in this research should be treated as to some extent under-estimated values."

***Comment*** [Methodology, part 3]:  Finally the 1D modelling explanation needs to be improved, especially in terms of boundary conditions. Please see the specific comments below.

 "The primary objective of the research was to present a methodology for generating FRM and PFM for the breaching of dike during a cyclone". This line from the conclusion should have similar versions in the abstract and introduction, and in the opinion of this reviewer, should shape the overall message of the paper. It is also closer to the title of the paper. Instead the abstract and introduction sections describe the paper as "an investigation of the inundation pattern in a protected area", which is an interesting case-study, but not innovative or of interest to a general audience. For example, a line of logic for the introduction could be;

- Bangladesh is susceptible to cyclones;
- Damage estimates and flood maps, as well the identification of critical breaching locations, are useful for mitigating risk
- Quantifying this data requires taking account of SLR, cyclone frequency etc. and potential for breaching.
- A method to do so for the 1:25yr event (design criteria), for a particular polder, is given below.

Then, as well as the specific results, the applicability of the method to other areas and return period events could be estimated.

***Reply***: *We appreciate this excellent suggestion and we have updated the manuscript to incorporate the suggestions. The specific changes in the manuscript are described below:*
*Abstract, P 1, Line 16 and 17:* "The primary objective of this research was to present a methodology for identifying the critical location of dike breach, generating flood risk map (FRM) and probabilistic flood map (PFM) for the breaching of dike during a cyclone."
*Introduction, P 2, Line 22 to 27:* "The coast of Bangladesh is frequently hit by severe cyclones (5 cyclones between 1995 and 2010, Dasgupta et al., 2014).  Bangladesh Water Development Board (BWDB) is responsible for the operation and maintenance of these dikes and lacks fund to conduct proper repairing of damaged dikes subsequent to any severe cyclone. As a result the dykes remain vulnerable to breaching. Identifying the critical location(s) of dike breach and prioritising the repairing of the critical location is likely to reduce the breaching possibility."
*P 3, Line 10 to 12:* "This paper presents a methodology to identify the critical location of dike breach due to cyclones, generate flood risk map (FRM) and probabilistic flood map (PFM) due to breaching of the dike by cyclone induced storm surges."
*P 3, Line 13 and 14:* "A cyclone event of 25 year return period was considered in this research. A coastal polder (Polder 48) of southern Bangladesh was selected as the study area."

***Comment***: **English:**
This has improved significantly, but again, continued errors distract the reader, especially in the conclusion. The paper cannot be accepted with so many mistakes, and it is strongly suggested to ensure they are removed before resubmission.
***Reply***: We have worked on improving the English of the manuscript, especially of the conclusions section. To the best of our understanding the language is now consistent across the manuscript.
***Specific Comments***:
***Comment***: **Abstract**

P1, Line 14: "Scenarios were developed by considering… geometrical properties of the breach, breach propagation time…". This line directly conflicts with P8, Line12: "As the geometry of the breach is not independent, it was not considered as a parameter for scenario development". This was highlighted previously and needs to be resolved.

*Reply: Thank you for pointing it out and we have updated manuscript as follows: P 1, Line 13 to 15):*

> "Scenarios were developed by considering tidal variations, angle of cyclone at landfall, possible dike breach locations, geometrical properties of the breach, breach propagation time and the sea level rise due to climate change according to the fifth assessment report (AR5) of Intergovernmental Panel on Climate Change (IPCC)."

*Comment*: **Introduction**

P2, Line 26: "Furthermore, by the year 2100 the annual estimated damage due to tropical cyclones may increase by US$53 billion" This is surely due to the reasons the authors mention previously, and therefore not 'furthermore'.

*Reply: Authors acknowledge the suggestion and the manuscript is adjusted accordingly.*

*P 3, Line 5 and 6:* "By the year 2100 the annual estimated damage due to tropical cyclones may increase by US$53 billion"

*Comment*: P2, Line 26: "study area is not benefitted with…" This sounds strange.

*Reply: Authors acknowledge the suggestion and the manuscript is adjusted accordingly (P 3, Line 7):*

> "At present, flood forecasting system is not available for the coastal region of Bangladesh."

*Comment*: **Study Area**

P3: Lines 9-13: This part seems just like a list of data, and doesn't connect well, consider revising.

*Reply: We have updated the manuscript based on this suggestion (P 3, Line 29 to P 4 Line 2):*

> "Climate and agricultural practices of Kuakata is similar to the climate and agricultural practices of the country (Bangladesh). The average yearly rainfall in Kuakata is 2590mm (Climate-Data, 2016) and the annual average temperature is 25.9°C (Climate-Data, 2016). Rabi (November to February), Kharif-I (March to May) and Kharif-II (June to October) are the three seasons for growing crops (DAE, 2009). The elevation of 80% of the area is 1.55m PWD, the vertical datum established by Public Works Department of Bangladesh, which is 0.46m below the MSL."

*Comment*: P4, figure 1: Please plot the extent of the dike on this map or on figure 3, as you mention it is only on the seaward side. Including the chainage points given in Figure 6 would also be useful.

*Reply: Thank you for the suggestion. As stated earlier in the section, study area (polder 48) is surrounded by dikes, the line denoting the study area also denotes the alignment of the dikes. This information is added to the manuscript to increase clarity. Chainage points and the extent of the dike for the area are added in the figure 3 (shown below as well).*

[Figure]

Figure 2: Schematic diagram of the study area with location of control structures and gauges and the considered breach locations.

***Comment*: Methodology**

***Comment***: P5, figure 2: The arrow from HEC-RAS to 'Estimate damage' during a loop is confusing. The output of 72 simulations from the loop should be used.

***Reply****: Authors thank the reviewer for the suggestion. The flow chart has been adjusted (Figure 2, page 5).*

***Comment***: P5, figure 2: Perhaps the damage and maps sections could be highlighted as outputs.

***Reply****: The flow chart has been adjusted following the suggestion indicating the estimated damage, probabilistic flood map and flood risk map as Output 1, 2 and 3 (Figure 2, page 5) respectively.*

***Comment***: P6, line 5: I don't think the HEC-Ras description here needs to be so detailed. In fact this whole paragraph should be reviewed for structure and cohesion.

***Reply****: The description of HEC-RAS was adjusted following the suggestion of the reviewer (P 6, Line 9 to 13):*

> "The river analysis tool HEC-RAS (version 5.0) from the US Army Corps of Engineers was used to develop the 1D-2D coupled inundation model. The flow in the river was modelled in 1D whereas the flow over the floodplain was modelled in 2D. HEC-RAS 5.0 is a free tool which can simulate 1D, 2D and 1D-2D coupled models for steady and unsteady flow. The 2D module of HEC-RAS provides option to simulate flow of water either with the diffusion wave equation or with the full shallow water equation (St. Venant equation). The availability of irregular flexible mesh in HEC-RAS and the option for faster simulations led to the selection of HEC-RAS 5.0 as the modelling tool."

***Comment***: P7, line 3-5: The English is poor here, and it is unclear. Are all cells rectangular? How can they have a single resolution or area? This is made more confusing by the grid representation in the figure.

***Reply****: Thank you for pointing this out. Authors acknowledge the suggestion and the manuscript was adjusted accordingly.*

*P 7, Line 6 to 7:* "The rectangular cells of the developed 2D mesh had a resolution of 25 m and the non-rectangular cells had areas ranging from 625 square meters to 1282 square meters."

**Comment**: P7: Figure 3: Presumably Khaprabhanga is more of a channel than a river, as it seems to connect two parts of the Bay of Bengal. If this is the case it should be mentioned.
**Reply**: *Khaprabhanga River is connected to Galachipa River on the east and Andharmanik River on the west. This is shown in Fig. 1.*

**Comment**: P7: Figure 3: How are the upstream and downstream locations selected? Presumably it is because it is assumed a cyclone always come from the west and therefore imposes a water level downstream, but this is never stated.
If I am correct in this assumption of a water level timeseries as the main driver in the model, why wasn't a QH relationship used at the other end? This raises a number of questions about the water level profile along the seafront.
Why was the Khaprabhanga included at all? No breaches occurred at that end, and it is not connected to the other 1D part.
**Reply**:
*We have updated Fig 3 to show western and eastern boundary instead of upstream and downstream boundary.*
*It was considered that the water level will be similar at all boundary locations. The upstream and downstream terminology was used to denote different boundary locations only. Authors agree that a simple terminology as boundary locations would have been enough.*
*Khapravabhga River was added to the model to make sure the results are similar to what happened during the cyclone Sidr when the dikes on the river side did not overtop or breached. The 1D network representing Khaprabhanga River was connected to the 2D model through lateral structures representing the dikes on the river side. The simulation of the storm surges during cyclone Sidr indicated that the dikes on the river side did not overtop indeed.*
**Comment**: P7, line 16: Mangrove ForestS.
**Reply**: *Authors acknowledge the suggestion and manuscript adjusted accordingly.*
*P 7, Line 18:*
"The western and eastern side of the embankment have Mangrove Forests between the rivers and the embankment"

**Comment**: P7: Line 17: "The storm surges…" English
**Reply**: *Authors acknowledge the suggestion and manuscript adjusted accordingly*

**Comment**: P8: Line 8: "…therefore, the wider and larger canals were included in the DEM". This sounds like the canals were 'burnt' into the DEM, which I understand was not the case.
**Reply**: *Indeed, the canals were not burned in the DEM. The DEM was generated from the land level survey points provided by IWM. These points included the points on the canals with bed level.*

**Comment**: P8: Line 8: "As the geometry of the breach is not independent, it was not considered as a parameter for scenario development". As mentioned above this is contradicts another statement. Also, the geometry is independent of the water levels, the failure is not.
**Reply**: *Authors thanks the reviewer for pointing this out. Authors acknowledge the suggestion and the manuscript was adjusted accordingly.*
*P 1, Line 13 to 15:*

"Scenarios were developed by considering tidal variations, angle of cyclone at landfall, possible dike breach locations, geometrical properties of the breach, breach propagation time and the sea level rise due to climate change according to the fifth assessment report (AR5) of Intergovernmental Panel on Climate Change (IPCC)."

**Comment**: P9, figure 6: Why is the middle chainage higher than the others during the rising limb? This raises more questions about the boundary conditions.
**Reply**: *Thank you for the query. But the authors failed to identify significant difference in water levels at the three chainage presented in figure 6.*

**Comment**: P10, Line 13: The 1:25yr design criteria should be mentioned in the abstract and introduction, as it dictates the method and outputs.
**Reply**: *Authors acknowledge the suggestion and manuscript adjusted accordingly*
*P 1, Line 15:*
"Storm surge for a cyclone event of 1 in 25 return period was considered for all the scenarios."
*P 2, Line 17 and 18:*
 "A storm surge event of 25 year return period was considered in this study for the generation of different scenarios."
*P 3, Line 14 and 15:*
"A cyclone event of 25 year return period was considered for this research."

**Comment**: P10, line 18: from which direction is the angle measured, i.e. does angle of landfall of 230 degress indicate a cyclone moving from South to North? Apologies if this is an obvious question, but I am not hugely familiar cyclones.
**Reply**: *It is measured from the south. The effect of the angle of landfall of the cyclone was studied by Azam et al., 2004. To illustrate how they defined the angle of landfall an example is provided below. In their article they defined the angle for the cyclone of 1991 shown in figure below as "During landfall, the cyclone made an angle of $215°$".*

[Figure]

Azam, M.H., Samad, M.A. and Mahboob-Ul-Kabir, 2004. Effect of cyclone track and landfall angle on the magnitude of storm surges along the coast of Bangladesh in the northern Bay of Bengal. *Coastal engineering journal*, *46*(03), pp.269-290.

**Comment**: P10, Line 34: "Single breach was considered for each scenario". English. Also, perhaps the authors can suggest the consequences of the assumption of 1 breach for a 1/25yr storm. Is 1 an over or under estimate? How many breaches have been recorded before?
**Reply**: *There are reports of dike breaching during a cyclone but no database is available for the location and number of the breach. Therefore, it is difficult to state if 1 breach is an underestimation or over estimation.*

**Comment**: P11, line 8; English

***Reply****: Authors acknowledge the suggestion and manuscript adjusted accordingly*

***Comment***: P11, line 10; It is important to clearly identify what the maps and outputs represent. For example the PFMS are probabilistic inundations maps for a single event (1/25yr) in which a breach occurs.
***Reply****: Authors acknowledge the suggestion and manuscript adjusted accordingly.*
*P 11, Line 12 to 13:*
"As the storm surge height suggested by Islam et al. (2013) corresponds to an event of 25 year return period, the PFM generated in this study corresponds to 1 in 25 year return period."

***Comment***: P12, line 23; The fact that you do not account for duration or velocity of inundation should be mentioned, especially if highlighting their importance.
***Reply****: The manuscript is adjusted following your suggestion.*
*P 12, Line 29 and 30:*
"The simplified (with regards to flow velocity and flood duration) depth-damage curve for rice fields suggested by Chau et al. (2014)"

***Comment***: P14, line 2. English
***Reply****: Authors acknowledge the suggestion and manuscript adjusted accordingly*

***Comment***: P14, line 7; "Pfj is the probability of reaching a certain storm surge level in simulation number j". Is each Pfj will be equal to 1/25 here? As mentioned I think it is important to specify these results relate to the 1/25yr level.
***Reply****: Authors thank the reviewer for the suggestion. Following the suggestion made by the reviewer in the previous comment the manuscript was adjusted accordingly.*
*P 11, Line 12 to 13:*
"As the storm surge height suggested by Islam et al. (2013) corresponds to an event of 25 year return period, the PFM generated in this study corresponds to 1 in 25 year return period."

***Comment***: **Results and Discussion**
P15, line 6: "…for identifying the critical location of breaching". This could be confused as a process to identify locations for the modelling process. Consider revising, and/or changing to 'regions'.
***Reply****: Authors' intention was to represent a method for comparing the results due to breaching and identify the most critical location of the dike for breaching. If these are identified and strengthened, then damage due to storm surges can be reduced significantly.*

***Comment***: P16, figure 9: Presumably these figures are averages of subsets of the 72 simulations, This should be indicated in the text.
***Reply****: figure 9 represented the flood extents of different scenarios considering the breaching at central part of the sea facing dike. The title of the figure was adjusted as "Comparison of flooded areas corresponding to different Scenarios considering the breaching at central part of the sea facing dike." to provide more clarity.*

***Comment***: P16, line 3: Now the authors move back to the subset of 3 scenarios. I find this hard to follow, especially since the names 'scenarios 1,2, and 3' do not relate to any cardinal numbers, but to breach locations on the west, centre and east.

***Reply****: Details about scenarios are presented in section "Cyclonic scenarios considered". Table 3 (Page 16) and Table 4 (Page 17) were adjusted for more clarity by adding western, central and eastern part to indicate which part of the dike was considered for breaching.*

***Comment*****:** P18, figure 11: This figure and figure 10 need clarification. Can I infer from them that the area in which the maximum flood depth was less than 0.1m in scenario 1 (western breach) had an inundation of just above 2km2, which causes about 400,000 euros of damage? I see this is later explained in the discussion, but the authors might consider a clearer explanation at this point too.
***Reply****: Authors acknowledge the suggestion and manuscript adjusted accordingly.*
P 18, Line 2 and 3:
"Figure 10 and Figure 11 corresponds to the flooded area and damage due to different range of inundation depth respectively. The flood area and damage was highest for the inundation depth of 0.5 m to 1.0 m."

***Comment*****:** P18, line 3: Section 4.5 describes how the western breach was selected as the critical location, but you should mention that result here, as the flood map is clearly dependent on that. Perhaps it should also be plotted in figure 12 as well.
***Reply****: Authors acknowledge the suggestion and manuscript adjusted accordingly.*
P 18, Line 5 and 6:
"Comparison of flooded area and damage due to flood for the three worst case scenarios lead to the identification of Scenario S1 as the critical location of breaching."

***Comment*****:** P19, line 8; "This arbitrary threshold…". It has just been explained that the threshold is not arbitrary.
***Reply****: Authors acknowledge the suggestion and manuscript adjusted accordingly*
P 19, Line 8:
"This threshold was used in developing the PMFs."

***Comment*****:** P19, line 8; "…and the estimation of damage due to flood was not affected". I don't understand this sentence at all.
***Reply****: Authors thanks the reviewer for pointing this out. The sentence indeed required more clarification and it was adjusted to provide clearer message.*
P 19, Line 8 and 9: "This threshold was not considered while the estimation of damage due to flood was conducted."

***Comment*****:** P20, figure 13: Again I think it should be mentioned that this is the 1/25yr PFM.
***Reply****: Following the previous suggestion, the manuscript was adjusted.*
P 11, Line 12 to 13:
"As the storm surge height suggested by Islam et al. (2013) corresponds to an event of 25 year return period, the PFM generated in this study corresponds to 1 in 25 year return period."

***Comment*****:** P20, line2: "Three worst case scenarios (Scenario 1, 2 and 3)…" Again there is confusion about these scenarios. They are not the 3 worst case scenarios, they are the worst case scenarios in terms of boundary conditions for each of the 3 potential breach locations.
***Reply****: Thank you for the suggestion and the manuscript is adjusted accordingly.*
P 11, Line 2 to 4:

"The highest storm surge height as the boundary condition with breaching at the western, central and eastern parts of the dike were considered as the worst case scenarios and were denoted as Scenario S1, S2 and S3 respectively."

**Comment**: P20, line 3: "demonstrate that large area is flooded…". English
*Reply: Authors thanks the reviewer, acknowledge the suggestion and manuscript adjusted accordingly*

**Comment**: P20, line 4: The text references figure 9, which relates to all 72 simulations, as I understand. This is in the middle of text about the 3 extreme scenarios, and further highlights the issue already mentioned.
*Reply: Authors acknowledge the suggestion and manuscript adjusted accordingly*
P 20, Line 2 to 4:
"The flood extent for the simulated result of 72 scenarios were compared. Flood extent varies for different scenarios with different conditions such as the daily (high and low tide) and biweekly (spring and neap tide) tidal variation, sea level rise and angle of landfall."

**Comment**: P20, line 5: "More than 25% of the total area of Polder 48 was inundated for the three scenarios...". Please make this sentence clearer, I understand it to mean 'At least 25% of the total area of Poder 48 was inundated in all 3 extreme scenarios"
*Reply: Authors acknowledge the suggestion and manuscript adjusted accordingly*
P 20, Line 7:
"At least 25% of the total area of Polder 48 was inundated for the three scenarios"

**Comment**: P20, line 10. It is not quite correct to say the risk is X million euro. Firstly it needs to be defined as a value per year or per event. Secondly these values are only the extreme versions of the 1/25yr event defined.
*Reply: Authors agree with the reviewer that the risk should not be defined in "X million euros". However, the line states the damage due to flood, not the flood risk.*

**Comment**: P21, line 8: English
*Reply*: Authors acknowledge the suggestion and manuscript adjusted accordingly

**Comment**: P21, line 10: "due to breaching of unprotected location of the dike". English
*Reply*: Authors acknowledge the suggestion and manuscript adjusted accordingly

**Comment**: P21, line12: "Scenario 1 had higher risk of flooding as the damage due to flooding was maximum for Scenario 1". This sounds strange, consider revising.
*Reply: Authors acknowledge the suggestion and manuscript adjusted accordingly*
P 21, Line 15 and 16:
"The damage due to flooding was maximum for Scenario S1 which results in higher risk of flooding for scenario S1."

**Comment**: **Conclusion**
P22, line 1: "investigate inundation pattern in a polder". English
*Reply*: Authors acknowledge the suggestion and manuscript adjusted accordingly

**Comment**: P22, line 2: "were compared using total flooded area and estimated damage". For which scenario set? 72 or 3?
*Reply: Authors acknowledge the suggestion and manuscript adjusted accordingly.*

"Different scenarios were formulated and simulated using a 1D-2D coupled model. The results of these simulations were used to calculate the total flooded area and damage due to flooding. Simulated results of three worst case scenarios S1, S2 and S3 were compared based on the total flooded area and estimated damage."

**Comment:** P22, line 5: "Flood risk map and probabilistic flood map…". English. These mistakes are very distracting.
**Reply:** Authors acknowledge the suggestion and manuscript adjusted accordingly

**Comment:** P22, line 8: "during the breaching of central part of the sea facing dike". It doesn't happen during the breaching.
**Reply**: *Authors acknowledge the suggestion and manuscript adjusted accordingly.*
*P 22, Line 20 and 21:*
"Flood inundation for the three worst case scenarios S1, S2 and S3 indicated that the maximum flooded area was obtained corresponding to the breaching of the central part of the sea facing dike."

**Comment:** P23, line 8: "Most flooded area had flood depth of 0.1 to 0.5 m…". English.
**Reply**: *Authors acknowledge the suggestion and manuscript adjusted accordingly.*

**Comment:** P22, line 20: "…are least and most probable land use respectively". This sounds strange.
**Reply**: *Authors acknowledge the suggestion and manuscript adjusted accordingly.*
*P 23, Line 5 and 6:*
"Inundation maps of all 72 scenarios were compared to generate the probabilistic flood map, which indicated that the areas with rice fields are the least and the settlements are the most probable areas to be flooded."

**Comment:** P22, line 24: This is the first mention of 'synthetic water level time series'. It is clear from the bottom of figure 6 that they are present, but not how they were generated.
**Reply**: The water levels presented in Figure 6 are result of having synthetic water level time series at the boundaries of the model. Unfortunately, measured water level during a cyclone was not available. The time series boundary file with the storm surge height considered for the 72 scenarios were generated following the measured tidal water level pattern. The following was added to clarify:
*P 8, Line 3 and 4:*
"The synthetic water level data for boundaries of the model were generated by following the tidal water level pattern and the storm surge height considered for the all scenarios."

**Comment:** P22, line 27:" This stresses the importance of field observation pre, post and during event". I don' see what actually stresses this importance.
**Reply**: Authors intended to state that due to lack of field observation, calibration and validation was difficult. Field observation will assist in future studies.

**Comment:** P22, line 30: "Comprehensive survey should be conducted…". English.
**Reply**: Authors acknowledge the suggestion and manuscript adjusted accordingly.

**Comment:** P22, line 33 "…a join probability" English
**Reply**: Authors acknowledge the suggestion and manuscript adjusted accordingly.

***Comment*:** P23, line 1: "Single breach was considered for all the developed Scenarios".
English
***Reply***: Authors acknowledge the suggestion and manuscript adjusted accordingly.

***Comment*:** P23, line 2: "…should be studies as well". English
***Reply****: Authors acknowledge the suggestion and manuscript adjusted accordingly.*

***Comment*:** P23, line 7: "Conducting field survey to generate these curves…" English
***Reply***: Authors acknowledge the suggestion and manuscript adjusted accordingly.
The authors are very thankful to the reviewer for the valuable comments. Addressing the comments has tremendously helped us in improving the manuscript.

**REVIEW COMMENTS OF REVIEWER # 2**

**Second review of manuscript:** "Flood risk assessment due to cyclone induced dike breaching on coastal areas of Bangladesh".

*Comment***: Overview and general remarks**

The second version of the manuscript is better than the first one. The authors put obvious effort in editing the text after the referee´s comments and suggestions. There are significant improvements in the Methods and Discussion/Conclusions section in terms of the content required for the study and readability. The Figures and Tables are clear. Most of language mistakes are corrected. Nevertheless, there are some improvements should be done, mostly related to refining some description in the Methods and Conclusion sections, missing/wrong reference and language. Therefore, I suggest to return the manuscript for minor revisions.

*Reply*: *We are thankful to the reviewer for the appreciation and in the following we provide answers to the specific review comments.*

*Comment***: Specific remarks**

Figure 12 and 13 miss a scale bar.

*Reply*: Authors acknowledge the suggestion and the figures were adjusted accordingly. The adjusted figures are presented below:

[Figure]

Figure 12: Flood risk map corresponding to the dike breach at the critical location of the dike. The following three classes of risk are shown: high, medium and low. The considered four landuses are shown as well.

[Figure]

Figure 13: Probabilistic flood map of the study area. Varying colours indicate probabilities of obtaining flood depths more than 0.5m.

**Comment:** p.2 line 10. Were *recently* designed.
*Reply: Thank you for pointing it out. The manuscript was adjusted accordingly.*
P 2, Line 16:
"Crest levels of the coastal dikes were recently designed for an event of 25 year return period"

**Comment:** p.2 line 20. The intensity of tropical cyclones is *likely* to increase.
*Reply: The line was adjusted as "Furthermore, the intensity and frequency of these tropical cyclones are likely to increase in the future due to climate change." (Page 2, Line 32)*

**Comment:** p.3 line 7. The land use is mainly ….. this sentence needs to be rephrased.
*Reply: The line was adjusted as "The land use is classified by the Ministry of Land, Bangladesh into the following four classes: rice fields, settlements, shrimp ponds and water bodies (river/canal)". (Page 3, Line 28 and 29)*

**Comment:** p.5 line 8. …(a company from Finland). In my opinion this information is unnecessary.
*Reply: Thank you for the suggestion and adjusted accordingly by removing the redundant information. (Page 5, Line 8)*

**Comment:** p.6 line 1. As was pointed out in the previous round of reviews, "model development" brings ambiguity to what is described. In the majority of cases "model development" means you worked and modified the source code of HEC-Ras in a certain way, if this was not the case I would recommend to change it to "set-up" or "build".
*Reply: Authors' acknowledge the suggestion for clarity and manuscript is adjusted accordingly by stating as "Setting up of 1D-2D coupled model".*

**Comment:** p.6 line 7. "Computationally efficient" needs references on other studies which prove the point.
*Reply: Authors' thanks the reviewer for the suggestion and line has been removed.*

**Comment:** p.12 line 18. Information missing on satellite image source and the way it was processed.
*Reply: Authors' acknowledge the suggestion for clarity and manuscript is adjusted accordingly.*
Page 12, Line 22 to 24:
"Satellite image from google earth was used for analysis. The satellite image of the area was downloaded and georeferenced. Then, the areas for buildings and open areas for households were manually calculated using ARCGIS."

**Comment:** p.12 line 25. "However, *when* the flood level …"
*Reply: Thank you for the suggestion and the line was adjusted accordingly following reviewer's suggestion. (Page 12, Line 32)*

**Comment:** p.12 line 30. "Further research …" such statements should be moved to the Conclusion section.

*Reply*: *Authors' acknowledge the suggestion for clarity and manuscript is adjusted accordingly by removing the statement.*

**Comment:** p.13 line 8. The reference (Helm, 1996) doesn't belong to the statement. The study of Helm gives the definition of the risk used in this manuscript at line 11. Klijn (2009) and Van Manen and Brinkhuis (2005) used already existing definitions for their research. This has to be corrected.
*Reply*: *Authors' acknowledge the suggestion and manuscript is adjusted accordingly by adjusting the references following the reviewer's suggestion (Page 13, Line 11 and Line 15)*

**Comment:** p.14 line 27. The indicator of error has to be specified.
*Reply*: *The line was adjusted as "The difference between the reported and the simulated flood depth was 4.5%" (Page 14, Line 29).*

**Comment:** p.20 line 17. "poor" is a rather vague definition, some numbers or indicators would bring more clarity.
*Reply*: *Authors' acknowledge the suggestion and manuscript is adjusted accordingly with numbers and references. (Page 21, Line 1 and 2).*

**Comment:** p. 21 line 2. "risky" is recommended to be removed.
*Reply*: *Authors' acknowledge the suggestion and manuscript is adjusted accordingly by adjusting the sentence as "Fishing in the coastal region of Bangladesh as it yields lower economic returns leading to enhanced poverty". (Page 21, Line 5 and 6).*

**Comment:** p.22 line 20. "Probabilistic flood maps…" is not clear, recommended to be rephrased.
*Reply*: *The line was adjusted as "Inundation maps of all 72 scenarios were compared to generate the probabilistic flood map, which indicated that the areas with rice fields are the least and the settlements are the most probable areas to be flooded." (Page 23, Line 5 and 6)*

**Comment:** p.23 line 5 "…, the data provided…" this part of sentence should be rephrased.
*Reply*: *Authors' acknowledge the suggestion and manuscript is adjusted accordingly as "Bathymetric data with coarse grid resolution from GEBCO was used as the measured bathymetric data for the sea was not available." (Page 23, Line 23 and 24).*

The authors are very thankful to the reviewer for the valuable comments. Addressing the comments has tremendously helped us in improving the manuscript.

**Round #3**

**RESPONSE TO REVIEWER'S COMMENTS**

**Comment (Overview):**
The paper has further improved from the previous version, and provides a clear explanation of its methodology and case results. The abstract, introduction and conclusion also now ensure the work is given better context for international readers. The authors should also be commended for their thorough responses to the issues raised, which makes subsequent reviews much less demanding. The general comments I had previously made (methodology, message and English) have all been addressed, as well as all the specific comments. I therefore suggest the paper to be 'accepted subject to minor revisions', suggestions for which are given below. While most are suggestions or very minor mistakes, the issue with the HEC-RAS boundary conditions is important and should be considered.
**Reply:** *We are thankful to the reviewer for the appreciation and in the following we provide answers to the specific review comments.*

**Abstract**
**Comment:** P1, Line 25: CycloneS
**Reply:** *Authors acknowledge the suggestion and the manuscript has been adjusted accordingly.*

**Comment:** P1, Line 27: LocationS
**Reply:** *Authors acknowledge the suggestion and the manuscript has been adjusted accordingly.*

**Introduction**
**Comment:** P3, Line 15-20: This seems to be a repeat of the abstract. Consider reducing or rewriting.
**Reply:** *Authors acknowledge the suggestion and the manuscript has been adjusted:*
"Resource intensive adjustment of the protective structures for the deltas around the world will be required as the frequency and intensity of the cyclone will increase with climate change. Along with structural measures, non-structural tools such as land use zoning with the help of flood risk maps and probabilistic flood maps has the potential to reduce risk and damage. Identification of critical location of breaching and intensifying the maintained effort for these locations can potentially prevent a disaster. The method presented in this research can be utilized for the vulnerable deltas around the world even though the coastal region of Bangladesh was selected as the case study area."

**Study Area**
**Comment:** P2, Line 1: 80% of the polder is at 1.55m? I assume the authors mean 80% is above/below 1.55m or that 80% of the polder has an elevation of about 1.55m. Please clarify.
**Reply:** *Authors thank the reviewer for pointing this out and the manuscript has been adjusted:*
"The elevation of 80% of the area is below 1.55m PWD"

**Comment:** P2, Line 2: MSL: the term has not been used before, please add (mean sea level)

*Reply:* *Authors thank the reviewer for pointing this out and the manuscript has been adjusted accordingly.*

**Comment:** P2, Line 6: Do these figures relate to this polder specifically? I.e. did 94 people die in this polder?
*Reply:* *Authors thank the reviewer for pointing this out. The numbers related death and damage to crops were for the Kalapara sub-district. Polder 48 (the study area) is within the administrative zone of Kalapara sub-district as stated in the "Study Area" section. The manuscript is adjusted as:*
"during cyclone Sidr, 94 people died and 45% of the crops were lost in Kalapara sub-district"

**Methodology**
**Comment:** P7, line 3: "…using discharge as the west boundary and water level as the east boundary conditions". Why is this done? It seems that the sea level (i.e. water level boundary
condition) will dominate the entire stretch on both 1D sections, and using Q as a BC just complicates things.
*Reply:* *Flow hydrograph at West and stage hydrograph at East was chosen. However, stage hydrograph at both ends could have been chosen as well.*

**Comment:** P7, Line 5: "HEC- 5 RAS generates mesh with irregular shapes." I think this can be removed, or changed to 'meshes'.
*Reply:* *We acknowledge the suggestion and the manuscript has been adjusted as:*
"HEC-RAS generates meshes with irregular shapes."

**Comment:** P13, line 4: "The adapted depth-damage curves are obviously simplistic ones.". I don't think this is needed.
*Reply:* *Authors thank the reviewer for the suggestion and the sentence has been removed.*

**Conclusions**
**Comment:** P22 Lines 22 – 26: For me there is no need to use the values here, they are already explained above.
*Reply: We agree with the reviewer and the manuscript has been modified accordingly.*

**Comment:** P22 Line 23: Comma and full stop after equal
*Reply:* *We acknowledge the correction and the manuscript was adjusted accordingly.*

*The authors are very thankful to the reviewer for the review comments. Addressing the review comments has certainly improved the quality of the manuscript.*

**List of relevant changes**

- *The revised manuscript provides more detailed information in the methodology section, by adding a flow chart to bring more clarity, following your suggestion. The methodology section has been improved in adding explanations on data, model development, scenario development and maps.*

- *All the chapters were adjusted in the new submitted manuscript to make the manuscript more coherent and connected to the sections of the paper.*

- *To improve the level of English, native English speakers have been consulted and the revised version of the manuscript has been adjusted following their suggestion.*

- *The main innovation of the research was presented clearly in the revised manuscript (The introduction and Conclusion section have been adjusted to present the research questions and innovations)*

- *The authors' intention was to present a global methodology applicable to everywhere. The relative sea level rise for RCP 8.5 of IPCC AR5 for the coast of Bangladesh is 0.56 m (GERIC, 2015) which is slightly lower than what was considered by the authors (0.63 m).*

- *The discussion section was adjusted with the reflection on the research questions. Limitations of the study were added in the conclusion (please see "Discussion" and "Conclusion" section).*

- *Muktadir and Hasan, 1985 stated in their study that the rural house hold of Bangladesh usually are built around a large country yard. The land classification by the Ministry of Land defines the whole house hold including the country yard as residential area. But the damage curve considered for the residential area was for the damage to the house and the country yard will not have significant damage if flooded. Considering this the authors tried to exclude the country yard from the residential area. The satellite image of the area indicated that the about half of the area of residential complex (house hold) is usually empty. Therefore, the figure 50% was considered. This was added in the revised version.*

- *The developed damage curves suggest that the damage for flood depth below 0.5 m is minimal. Moreover, the authors tried to explore the effect of living with flood concept. Also tried to consider the uncertainty of the DEM and the 2D inundation model. The depth as 0.5m is an arbitrary depth. For a country where flood is a recurrent phenomenon with larger depth, living with flood might be already adopted by the local people. Moreover, this arbitrary was used for PMFs and the estimation of damage due to flood was not affected. This was added in the revised manuscript.*

- *The manuscript was updated to reflect the applicability at other location.*

[revised manuscript text omitted]

- *Study area (polder 48) is surrounded by dikes, the line denoting the study area also denotes the alignment of the dikes. This information is added to the manuscript to increase clarity. Chainage points and the extent of the dike for the area are added in the figure 3 (shown below as well).*

[Figure]

- *The description of HEC-RAS was adjusted following the suggestion of the reviewer (P 6, Line 9 to 13):*
  "The river analysis tool HEC-RAS (version 5.0) from the US Army Corps of Engineers was used to develop the 1D-2D coupled inundation model. The flow in the river was modelled in 1D whereas the flow over the floodplain was modelled in 2D. HEC-RAS 5.0 is a free tool which can simulate 1D, 2D and 1D-2D coupled models for steady and unsteady flow. The 2D module of HEC-RAS provides option to simulate flow of water either with the diffusion wave equation or with the full shallow water equation (St. Venant equation). The availability of irregular flexible mesh in HEC-RAS and the option for faster simulations led to the selection of HEC-RAS 5.0 as the modelling tool."

- *Fig 3 was updated to show western and eastern boundary instead of upstream and downstream boundary.*

- *The following changes were made in the manuscript for clarity*

*P 1, Line 15:*

[revised manuscript text omitted]

The 1D section of the model was initially developed and calibrated using the information shared by IWM. The 1D part of the developed model was calibrated for non-floodnormal conditions as measured discharge/ water level data during a cyclone event was unavailable. The model was simulated using discharge as the westupstream boundary and water level as the

5 eastdownstream boundary conditions (Figure 3). The calibrated 1D model was then coupled with the 2D model of flow over the floodplain using the a Digital Elevation Model (DEM) of the study area to include the 2D component and overland flow. The DEM was generated from the land level survey conducted by IWM and FINMAP. The land level survey of IWM did not cover the whole study area. The gaps in land level survey of IWM were filled in with the data from a previous survey conducted by FINMAP, which was provided by IWM. FINMAP (a company from Finland) conducted the topographic surveys in 1988

10 (MIWF, 1993). The elevation differences of land survey of IWM (conducted in 2012) and FINMAP indicated the land subsidence. The FINMAP data was corrected for land subsidence using the arithmetic average of the differences of these two land level surveys for Polder 48. The combined DEM has a resolution of 50 m.
The bathymetric data for the sea was collected from global bathymetric chart of ocean (GEBCO) (Smith and Sandwell, 1997). The land use data was collected from the Ministry of Land of Bangladesh. MODIS reflectance data was used for the analysis

15 of previous flood events. The methodology and equations suggested by Hoque et al. (2007) were used to analyse the MODIS reflectance data to determine flood extents during previous flood events. The intention was to utilise the flood extent generated from MODIS reflectance for calibration. But the MODIS data during the event was not available and the analysis of earliest available image after the cyclone event did not indicate of flooding.

[revised manuscript text omitted]